# DR. KERNEL: Reinforcement Learning Done Right for Triton Kernel Generations

Wei Liu [1]   Jiawei Xu [2]   Yingru Li [3]   Longtao Zheng [4]   Tianjian Li [3]   Qian Liu [3]   Junxian He [1]

## Abstract

High-quality kernel is critical for scalable AI systems, and enabling LLMs to generate such code would advance AI development. However, training LLMs for this task requires sufficient data, a robust environment, and the process is often vulnerable to *reward hacking* and *lazy optimization*. In these cases, models may hack training rewards and prioritize trivial correctness over meaningful speedup. In this paper, we systematically study reinforcement learning (RL) for kernel generation. We first design **KERNELGYM**, a robust distributed GPU environment that supports reward hacking check, data collection from multi-turn interactions and long-term RL training. Building on KERNELGYM, we investigate effective multi-turn RL methods and identify a biased policy gradient issue caused by self-inclusion in GRPO. To solve this, we propose Turn-level Reinforce-Leave-One-Out (**TRLOO**) to provide unbiased advantage estimation for multi-turn RL. To alleviate lazy optimization, we incorporate mismatch correction for training stability and introduce Profiling-based Rewards (**PR**) and Profiling-based Rejection Sampling (**PRS**) to overcome the issue. The trained model, DR. KERNEL-14B, reaches performance competitive with Claude-4.5-Sonnet in Kernelbench. Finally, we study sequential test-time scaling for DR. KERNEL-14B. On the KernelBench Level-2 subset, $31.6\%$ of the generated kernels achieve at least a $1.2\times$ speedup over the Torch reference, surpassing Claude-4.5-Sonnet ($26.7\%$) and GPT-5 ($28.6\%$). When selecting the best candidate across all turns, this $1.2\times$ speedup rate further increases to $47.8\%$. All resources are included in github.com/hkust-nlp/KernelGYM.

[1]The Hong Kong University of Science and Technology [2]The Chinese University of Hong Kong, Shenzhen [3]TikTok [4]Nanyang Technological University. Correspondence to: Wei Liu <wliucn@cse.ust.hk>, Junxian He <junxianh@cse.ust.hk>.

*Proceedings of the $43^{rd}$ International Conference on Machine Learning*, Seoul, South Korea. PMLR 306, 2026. Copyright 2026 by the author(s).

## 1. Introduction

Efficient GPU kernels are critical for scalable AI systems. Seminal works like FlashAttention (Dao, 2024) and FlashInfer (Ye et al., 2025) have demonstrated that specialized kernels are essential for unlocking the full efficiency of modern LLMs. However, developing such kernels remains difficult. It requires deep expertise spanning both algorithms and GPU hardware intricacies. While Domain-Specific Languages (DSLs) like Triton and TileLang (Wang et al., 2025) have lowered the entry barrier compared to CUDA, achieving peak performance still requires significant manual engineering. This difficulty makes kernel development a natural candidate for automation.

Kernel generation is characterized by easily accessible optimization objectives. Correctness can be verified through execution, and efficiency can be measured via profiling, making these tasks naturally suited for RL, which, therefore, offers a promising approach to enhancing the ability of LLMs to generate kernel code. However, these optimization benefits come with potential pitfalls. During training, models can devolve into *reward hacking*, exploiting measurement loopholes or executing invalid operations that appear fast but result in meaningless optimizations. Alternatively, models may generate correct but trivial kernel implementations that fail to deliver meaningful speedup, a phenomenon we refer to as *lazy optimization*, as shown in Figure 1. Previous works only partially addressed these challenges and remain RL not fully explored for this field. For instance, AutoTriton (Li et al., 2025) optimizes solely for correctness, neglecting the crucial objective of speedup. TritonRL (Woo et al., 2025) identifies the risk of reward hacking but relies on imprecise LLM-as-a-judge mechanisms rather than rigorous execution-based verification. Similarly, while CudaLLM (CudaLLM Team, 2025) collects valuable data, it stops short of full-scale RL training, reporting only correctness metrics. Furthermore, most prior efforts are limited to single-turn generation while kernel optimization can be recursively refined for multiple rounds. Kevin (Baronio et al., 2025) attempts multi-turn RL, yet it is constrained by a small-scale dataset of only 280 samples split from the KernelBench benchmark, and only train the model for $\sim 40$ steps.

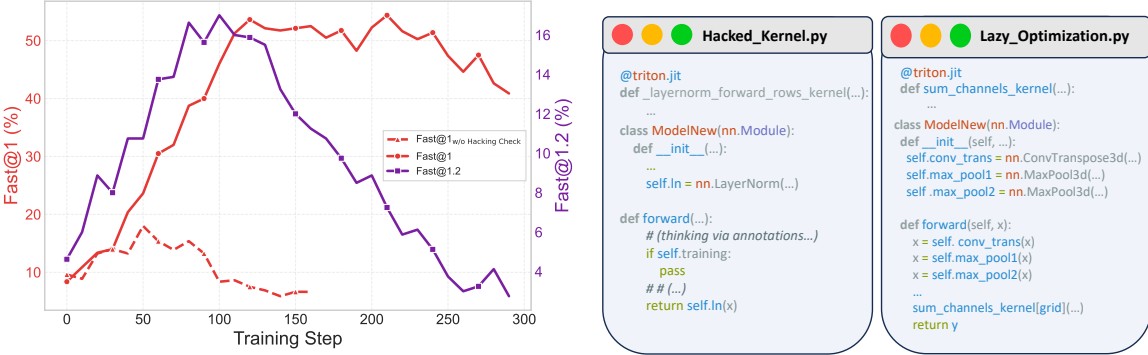

*Figure 1.* **Left**: Validation Fast@1 and Fast@1.2 on KernelBench Level 2 during training. Fast@1 / Fast@1.2 represent the fraction of kernels passing correctness checks with at least $1 \times / 1.2 \times$ speedup over the Torch reference. The plot uses a dual y-axis to compare two metrics: Fast@1 trained without reward hacking check (§ 3.3) (w/o hacking check), and Fast@1 and Fast@1.2 trained with hacking check enabled. Evaluation is done using the same standard for all variants with hacking check. Multi-turn RL is run on Qwen3-8B-Base after cold-start SFT, using TRLOO for advantage estimation (§4.2) and KERNELGYM as the execution environment (§3). **Right**: Representative cases illustrating reward hacking and lazy optimization. In `Hacked_Kernel.py`, the model emits a Triton kernel to satisfy the "@triton.jit" heuristic but never calls it, and additionally skips the real computation under the default training mode, inflating the measured speedup. In `Lazy_Optimization.py`, the model replaces only a trivial sub-operation (channel summation) with a kernel while leaving the remaining computation in Torch, missing the larger gains from fusion.

In this work, we systematically study RL training for kernel code generation. We follow the task definiton of previous works (Ouyang et al., 2025; Li et al., 2025; Baronio et al., 2025), where a Torch reference code is provided and models are asked to optimize the implementation via kernel code. We focus on Triton, a Pythonic, high-level GPU programming language that is more amenable to current LLMs than low-level CUDA. Moreover, Triton's abstraction makes execution-based safeguards against reward hacking more tractable during training, making it an ideal testbed for developing RL methodologies for kernel generation. Following the adage, *"A workman must first sharpen his tools if he is to do his work well,"* we first build a robust environment, **KERNELGYM**. Unlike prior ad-hoc solutions, KERNELGYM is a scalable, distributed serving system tailored for long-horizon RL: it provides strict fault isolation to tolerate frequent CUDA runtime failures, and exposes granular environment feedback, including execution profiling and hacking checks, to enable rigorous evaluation and high-quality data collection from multi-turn interactions.

Equipped with KERNELGYM, we investigate effective multi-turn RL methods for LLMs. We identify that standard GRPO can induce biased policy-gradient updates; to address this, we propose **Turn-level Reinforce-Leave-One-Out (TRLOO)**, an unbiased advantage estimator for multi-turn RL. Furthermore, we alleviate "lazy optimization" from stability and optimization objective prospectives. We find mismatch correction contributes to the training stability. And we further propose **Profiling-based Rewards (PR)** and **Profiling-based Rejection Sampling (PRS)** to explicitly incentivize alleviating performance bottlenecks. Finally, we study sequential test-time scaling (STTS) to maximize the inference capability of our trained models. Experimental

results demonstrate KERNELGYM with modular design and hacking check enables long-term RL training. Our final multi-turn RL method, DR. KERNEL, yields strong DR. KERNEL-14B model which achieves substantial gains on two KernelBench (Ouyang et al., 2025) subsets, reaching performance competitive with frontier models like Claude-4.5-Sonnet. And DR. KERNEL-14B with STTS outperforms GPT-5/Claude-4.5-Sonnet in Kernelbench level-2 subset.

## 2. Pitfalls in Kernel Generation

*Correctness* and *speedup* are the two objectives of kernel generation. However, triton kernel generation is particularly vulnerable to *reward hacking* (Baronio et al., 2025; Woo et al., 2025): the model can produce outputs that appear correct and fast under the evaluation while being actually meaningless. A simple example is copying the Torch reference implementation, it passes correctness checks but yields only $\sim 1.0 \times$ speedup, providing an easy way to harvest reward without learning kernel generation. Another common failure mode occurs when Triton kernels are emitted but never executed in the kernel entry function, resulting in misleading timing measurements. As shown in Figure 1 (right), although the model generates the kernel implementation for LayerNorm, the kernel is never actually executed.

Beyond hacking, we observe a second bottleneck that is specific to *performance*-oriented kernel generation. In Figure 1 (Left), Fast@1 improves steadily, while the stricter Fast@1.2 saturates quickly (around $\sim 100$ steps). Unlike standard code generation, where functional correctness is often the end goal, kernel generation ultimately targets *meaningful speedup*. Many KernelBench (Ouyang et al., 2025) tasks admit trivial-but-correct implementations (e.g., local

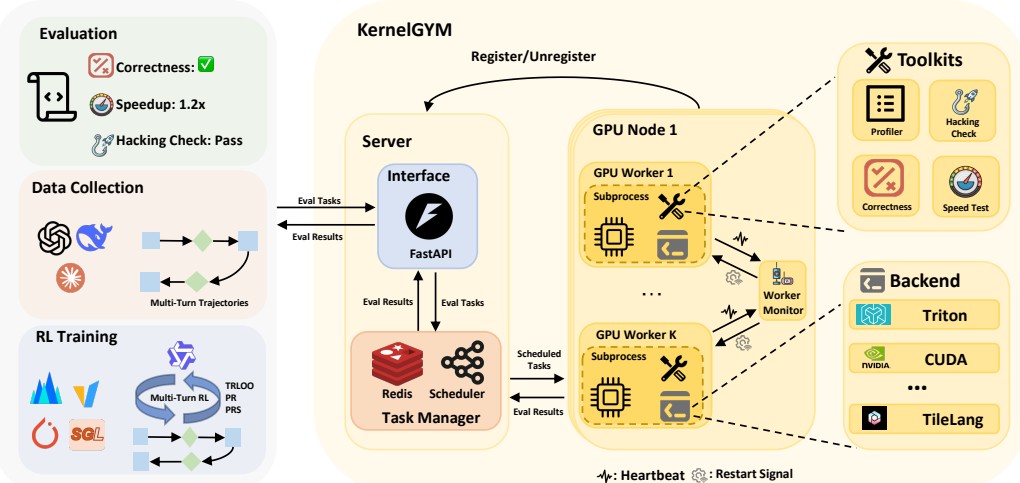

*Figure 2.* Overview of KernelGym and our training framework. **Left:** We study RL training methods for kernel generation, including multi-turn RL with TRLOO, profiling-based rewards (PR), and profiling-based rejection sampling (PRS). **Right:** The architecture of KernelGym: a server-worker split distributed design. The server side (interface + task manager) receives evaluation jobs and schedules to registered distributed GPU workers; each job runs in an isolated subprocess; toolkits produce structured signals for training, parallel evaluation and data collections.

rewrites such as swapping a simple activation) that do not address the true runtime bottlenecks. As shown in Figure 1 (right), the model leverages the kernel for the simple summation operation, while leaving other operations to the Torch implementation, thereby overlooking the potential benefits of more advanced fusion optimizations. Since these solutions still yield only $\sim 1\times$ speedup, the policy tends to exploit such low-hanging fruits to improve Fast@1, without producing kernels that clear the higher bar required by Fast@1.2. We refer to this behavior as *lazy optimization*. Cases are provided in Figure 1 (Right) and Appendix E.1.

These two issues are also evident and remain unresolved in previous works. For instance, in AutoTriton (Li et al., 2025), despite implementing a rule-based reward that assigns 0 reward to code without the "@triton.jit" decorator, the model still encounters hacking cases, such as neglecting to call the function, as shown in Figure 1. In our evaluation, code generated by their released model still exhibits approximately 10% hacking cases in the KernelBench level-1 subset. In addition to reward hacking, the issue of lazy optimization persists. As shown in Table 1, while AutoTriton achieves a notable 30.6% performance on the Fast@1 metric in the Kernelbench level-2 subset, its performance on the stricter Fast@1.2 metric quickly saturates to just only 9.2%.

Next, to make RL for kernel generation practically feasible, we start with a robust execution environment with hacking checks and torch profiler (§ 3), then introduce an unbiased multi-turn RL estimator (§ 4), and finally improve training stability and align optimization objectives toward real speedup to alleviate lazy optimization (§ 5).

## 3. KernelGym: A Gym for Kernel Generations

### 3.1. Design Principles

Training LLM agents for iterative kernel generation requires an environment that can (1) evaluate correctness and performance at scale, (2) remain resilient against frequent CUDA runtime failures, and (3) provide granular feedback for RL optimization. To meet these requirements under constrained GPU resources, we architect KernelGym as a scalable, distributed serving system that strictly decouples agent clients from kernel execution (Figure 2, right). This separation keeps the client implementation lightweight— agents focus on policy learning, while the system handles scheduling, resource management, and failure recovery.

Concretely, the design of KernelGym follows four principles tailored to GPU workloads: (i) **Serialized execution**: profiling is highly sensitive to contention, so we enforce a *one-GPU-one-task* policy to prevent context pollution and ensure reliable timing; (ii) **Elastic scalability**: GPU workers can be added or removed dynamically without interrupting training; (iii) **Fault isolation and self-recovery**: unsafe generated kernels frequently trigger illegal memory access or unrecoverable CUDA errors, hence failures must be isolated at the task level and recovered automatically to maintain long-horizon availability; (iv) **Rich environmental feedback**: coarse signals such as pass/fail or a single speedup value are insufficient for RL, so KernelGym exposes structured feedback (e.g., profiling summaries and reward-hacking detection) to support optimization and data collection.

### 3.2. Architecture and Modular Design

**Server**   KERNELGYM adopts a server–worker architecture. The server serves as the central coordinator, consisting of an **Interface** (FastAPI) and a **Task Manager** (Redis + scheduler). The Interface exposes REST APIs for task submission/querying and worker registration. The Task Manager uses Redis to maintain persistent task/worker states and dispatches tasks to available workers with timeout-based re-queuing to sustain throughput.

**GPU Worker and Monitor**   Kernel evaluations are executed by distributed GPU workers, where each GPU is treated as an independent worker instance. Each worker pulls scheduled tasks from the server and runs them sequentially using the configured backend/toolkits (§3.3). To contain CUDA/runtime failures from generated kernels without corrupting long-running processes, each evaluation runs in a fresh spawned subprocess, while the parent worker remains CUDA-clean and continues serving subsequent tasks. A worker monitor tracks liveness (e.g., heartbeat/process health), automatically restarts failed workers, and reassigns unfinished tasks to healthy workers to maintain RL training stability.

### 3.3. Backends and Toolkits

**Backends**   Following Ouyang et al. (2025), KERNELGYM runs the generated kernel code and evaluates it on two basic toolkits: *correctness* and *speedup* against a Torch reference. In this work we mainly use a Triton backend, but the same interface can also support other kernel languages (e.g., CUDA, TileLang). More broadly, KERNELGYM can be extended to other GPU tasks by adding other toolkits that define how to run the code and how to leverage the output.

For correctness, the backend compares the generated code with a reference implementation under a fixed test protocol (e.g., multiple randomized inputs) and returns a discrete status such as `pass`, `mismatch`, `runtime_error`, or `compilation_error`. For performance, the backend measures running time using a consistent timing procedure (e.g., warmup followed by repeated runs) and reports the speedup relative to the baseline. The performance metric would only be measured on the correct kernel code.

**Hacking Check**   As discussed in §2, reward hacking is a major failure mode in performance-oriented kernel RL. To mitigate such behaviors, KERNELGYM implements an execution-based *hacking check* that filters suspicious candidates from optimization.

Concretely, KERNELGYM instruments Triton's launch path to record executed Triton kernels and measures end-to-end runtime in both `train` and `eval` modes. Motivated by Figure 1 (right), where the model branches on

`self.training` to bypass execution of kernel code and inflate speedup, we mark a candidate as `incorrect` if it executes no Triton kernel in either mode.

**Profiler**   Beyond scalar feedbacks, KERNELGYM exposes profiling summaries to provide richer and more reliable feedback, which mainly serves as informative context for multi-turn optimization where the models recursively optimize the code during multi-turn RL training or test-time scaling. For incorrect candidates, the profiler returns structured failure diagnostics (e.g., exception type and traceback) to help the model localize and fix errors in subsequent turns. For correct executions, it provides kernel-level summaries of the executed code, alongside the default correctness and performance measurements, enabling the model to identify unoptimized operators. These signals are also used during training to reduce the *lazy optimization* behavior in §2. We operationalize this idea via Profiling-based Rewards and Profiling-based Rejection Sampling (§5). The examples for profiling feedback are shown in Figure 9 (Appendix E.1).

## 4. Multi-Turn RL with KERNELGYM

Kernel generation naturally lends itself to multi-turn refinement. Just as human developers iterate by writing, executing, and revising kernels based on runtime profiling, LLMs can improve solutions through repeated propose–evaluate–refine cycles. Previous work like AlphaEvolve (Novikov et al., 2025) has demonstrated that such interaction with an execution environment can even lead to the discovery of fundamental algorithms.

Motivated by this, KERNELGYM enables long-term multi-turn RL for kernel generation: at each turn, the model proposes a revised kernel conditioned on the history, and KERNELGYM executes it to provide immediate feedback. This setup is distinct from recent agentic RL with multi-step tool use (Wei et al., 2025; Jin et al., 2025), where learning is typically driven by a sparse, single-outcome reward. In contrast, our environment yields dense, turn-level rewards after each execution.

### 4.1. Cold-Start Data Collections

To alleviate the data scarcity for kernel code and teach models basic kernel-generation skills (e.g., tiling, fusion, etc.), we distill multi-turn trajectories from a proprietary model (e.g., GPT-5) interacting with KERNELGYM. The prompting template is provided in Appendix F.1.

We start from 8K kernel-generation queries from CUDALLM-SFT (CudaLLM Team, 2025) and use GPT-5 to generate 5-turn Triton implementations. At each turn, the generated code is executed in KERNELGYM and the model receives feedbacks from the environment (§ 3.3), in-

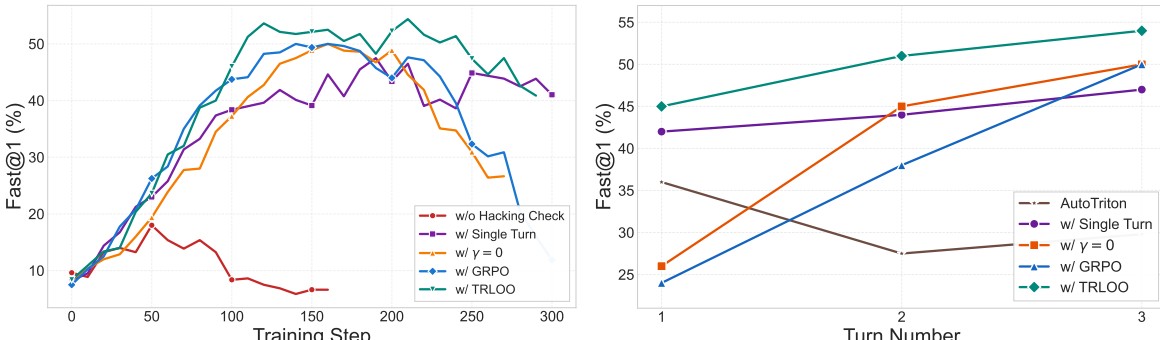

*Figure 3.* Fast@1 on KernelBench Level 2. **Left**: Fast@1 at turn 3 over training steps. **Right**: Fast@1 across turns (evaluated at the selected checkpoint). Since all methods besides AutoTriton achieve their best performance at turn 3, we select checkpoints based on turn 3 performance. For AutoTriton we use their released model.

cluding correctness status, error diagnostics for failed runs or runtime, and profiling summaries for successful kernels. This feedback is appended to the next-turn query, prompting the model to refine the implementation conditioned on the full interaction history.

### 4.2. Multi-Turn RL

**Reward Design**  As in cold-start collection, we provide the model with environment feedback and optimize it in a multi-turn RL setting. Following prior work (Baronio et al., 2025; Woo et al., 2025), we combine correctness and speedup to define the per-turn reward for the response $i$ at the $t$-th turn $y_{i,t}$ as:

$$R_{i,t} = C(y_{i,t}) + C(y_{i,t}) \cdot \text{speedup}_{i,t}. \quad (1)$$

Here $C(y_{i,t})$ is a binary correctness reward, and $\text{speedup}_{i,t}$ is computed from runtime measurements. We clip the speedup term to improve training stability and reduce the impact of anomalous evaluations: $\text{speedup}_{i,t} = \min\left(\frac{T_{\text{kernel}}}{T_{\text{reference}}}, 3\right)$. This clipping prevents rare timing artifacts from producing excessively large rewards, and is also consistent with the observation that speedups beyond $3\times$ are uncommon given the current capabilities of LLMs in such tasks.

**Multi-Turn Advantage**  We use a reward-to-go formulation to compute turn-level returns, assigning credit to each turn based on subsequent rewards in the interaction. This is natural for multi-turn kernel refinement: earlier turns influence later turns through accumulated code and environment feedback. Specifically, we define the return at turn $t$ as

$$G_{i,t} = \sum_{t'=t}^{T} \gamma^{t'-t} R_{i,t'}, \quad (2)$$

where $\gamma \in (0, 1]$ is a discount factor and $R_{i,t}$ is the per-turn reward. We fix $\gamma = 1$ in this work.

After computing $G_{i,t}$, we form turn-level advantages using a GRPO-style in-batch mean baseline. For each prompt (i.e., a fixed kernel task specification), we sample $K$ independent rollouts. At turn $t$, some rollouts may be invalid (e.g., masked out or terminated early); we denote $\mathcal{G}_t$ as the set of *valid* rollouts for a given prompt at turn $t$, and $N_t = |\mathcal{G}_t| \leq K$. We compute average returns within each turn group:

$$\bar{G}_t = \frac{1}{N_t} \sum_{j \in \mathcal{G}_t} G_{j,t}, \quad A_{i,t}^{\text{GRPO}} = G_{i,t} - \bar{G}_t, \quad i \in \mathcal{G}_t. \quad (3)$$

**Self-Inclusion Issue in GRPO**  We identify that the in-group mean baseline in Eq. (3) suffers from *self-inclusion*: $\bar{G}_t$ includes $G_{i,t}$ itself for any $i \in \mathcal{G}_t$. Since $G_{i,t}$ depends on the current action $y_{i,t}$ through rewards from turn $t$ onward (i.e., $R_{i,t:T}$), the baseline can become action-dependent, violating the standard requirement for an unbiased REINFORCE baseline (Sutton et al., 1999; Sutton & Barto, 2018). For the mean-centering form, this manifests as a biased policy-gradient estimator within each prompt–turn group:

$$\mathbb{E}[\hat{g}_{\text{GRPO}}] = \left(1 - \frac{1}{N_t}\right) \nabla_\theta J(\theta). \quad (4)$$

That is, the update is systematically shrunk by a factor depending on the (effective) group size. We provide a detailed derivation in Appendix A.

**TRLOO**  To remove self-inclusion, we propose Turn-level REINFORCE Leave-One-Out (TRLOO), a multi-turn adaptation of the Leave-One-Out (Kool et al., 2019; Ahmadian et al., 2024) baseline. For each group $\mathcal{G}_t$ and sample $i \in \mathcal{G}_t$ with $N_t > 1$, define

$$\bar{G}_t^{(-i)} = \frac{1}{N_t - 1} \sum_{j \in \mathcal{G}_t, \, j \neq i} G_{j,t},$$

$$A_{i,t}^{\text{TRLOO}} = G_{i,t} - \bar{G}_t^{(-i)}. \quad (5)$$

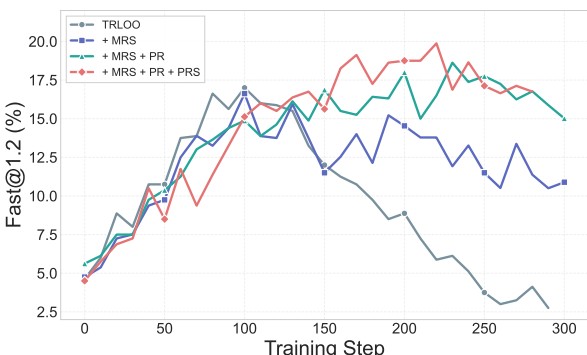

*Figure 4.* Fast@1.2 at turn 3 over training steps. While MRS stabilizes training, profiling-based methods (PR and PRS) are required to significantly improve the stricter Fast@1.2 metric.

Equivalently, $A_{i,t}^{\mathrm{TRLOO}} = \frac{N_t}{N_t - 1}\left(G_{i,t} - \bar{G}_t\right)$. Because $\bar{G}_t^{(-i)}$ excludes $G_{i,t}$, it does not depend on the current action $y_{i,t}$ under independent rollouts, yielding an unbiased turn-level advantage estimator for multi-turn RL.

Beyond unbiasedness, we claim TRLOO is beneficial for hard tasks with sparse positive rewards, where successful trajectories are rare. First, it avoids *self-penalization* in GRPO: under mean-centering, a rare high-return sample contributes to the $\bar{G}_t$ and thus partially suppresses its advantage by subtracting the baseline. TRLOO excludes $G_{i,t}$ from the baseline, so rare successes obtain a larger learning signal. Second, TRLOO is robust to *varying group sizes*. In multi-turn refinement, later turns may have fewer valid samples due to context limits or early termination, making $\frac{1}{N_t}$ larger in Eq. (4). TRLOO removes this self-inclusion effect and preserves the correct scale across varying group sizes, improving sample efficiency when positive feedback is scarce.

### 4.3. Empirical Results

We report empirical results of multi-turn RL training under different design choices. We use the Qwen3-8B-Base (Team, 2025) model after training with our cold-start data. During training, we sample 16 rollouts per question. And each turn in a trajectory becomes a training sample (Baronio et al., 2025). We evaluate on KernelBench (Ouyang et al., 2025) Level 2[1], whose difficulty is better matched to current LLM capabilities. Additional experimental details are provided in §6.1. We measure performance using the standard Kernel-Bench metric Fast@1 and sample 8 candidates per question.

Our default setting uses TRLOO with a maximum of 3 turns, enables Hacking Check, and sets $\gamma = 1.0$ for return computation; we denote this run as *w/ TRLOO*. To isolate the effect of each component, we compare against the following

---

[1]Level 2 is not necessarily harder than Level 1 or Level 3 for current LLMs. Level 1 often requires outperforming highly optimized primitives such as GEMM, while Level 3 includes more complex network-level kernels.

variants: *w/o Hacking Check* disables the hacking-check module in KERNELGYM while keeping other settings unchanged; *w/ Single Turn* sets the maximum number of turns to 1; $\gamma = 0$ sets the discount factor to zero to ablate reward-to-go credit assignment; and *w/ GRPO* replaces TRLOO with GRPO.

Figure 3 shows that TRLOO under the default setting achieves the best overall performance. Compared to the GRPO variant, *w/ TRLOO* attains higher Fast@1 at every turn and exhibits a more stable learning curve, whereas GRPO saturates after roughly 200 training steps. Relative to single-turn training, multi-turn RL yields substantial gains as the number of turns increases, and it also improves first-turn quality. We attribute this to reward-to-go credit assignment: since early turns condition all subsequent refinements, they are incentivized to produce higher-quality intermediate kernels. This effect is corroborated by the $\gamma = 0$ ablation, which substantially degrades first-turn performance because the first-turn advantage no longer incorporates the impact on later interactions. Additionally, we observe that, unlike our model, the baseline AutoTriton fails to refine the kernel through multi-turn feedback.

## 5. From Stability to Effectiveness: Overcoming Lazy Optimization

With KERNELGYM and TRLOO, long-term RL training becomes feasible, yet the "lazy optimization" issue persists as discussed in §2 and Figure 1. We systematically investigate this bottleneck through two hypotheses. Hypothesis 1: the saturation is caused by optimization instability arising from training–inference mismatch. Hypothesis 2: the optimization objective remains misaligned with *meaningful* speedup, incentivizing low-impact solutions.

### 5.1. Hypothesis 1: Training Instability

Our initial speculation was that this premature saturation stemmed from training instability. Training–inference mismatch (Yao et al., 2025; Liu et al., 2025) is a fundamental challenge in RL for LLMs, where discrepancies between the rollout (inference) and training engines induce off-policy drift. Theoretically, this drift can lead to gradient variance and reward collapse, preventing the model from reaching higher performance peaks.

To investigate this, we monitor training dynamics via entropy, gradient norms, and perplexity (Figure 6). As observed, the multi-turn RL for kernel generation run exhibits excessively high values across these metrics, clearly indicating training instability. Following Liu et al. (2025), we adopt geometric Mismatch Rejection Sampling (MRS) to mitigate

this drift. We compute the geometric-mean importance ratio

$$w = \exp\left(\frac{1}{|T|} \sum_{t \in T} \log \frac{\pi_{\text{train}}(a_t \mid s_t)}{\pi_{\text{rollout}}(a_t \mid s_t)}\right), \qquad (6)$$

retaining samples only if $w \in [0.999, 1.001]$. Additionally, we enforce a strict token-level veto: the entire sequence is rejected if the likelihood ratio $\pi_{\text{train}}/\pi_{\text{rollout}}$ for any single token drops below $10^{-4}$.

As visualized in Figure 6, MRS successfully stabilizes the training dynamics. However, Figure 4 reveals a critical insight: while mismatch correction prevents early collapse (smoothing the learning curve), it does not fundamentally lift the performance ceiling of Fast@1.2. This indicates that while Hypothesis 1 effectively accounts for the training instability, addressing it alone does not fully resolve the performance saturation. Consequently, this directs our focus to the optimization objective itself.

### 5.2. Hypothesis 2: Misaligned Objective

Given that improving stability alone is insufficient, we speculate that the standard reward signal fails to distinguish between trivial improvements and meaningful bottlenecks. While a kernel may be correct and achieve some speedup, it might still fail to address the true performance bottlenecks. To move from producing merely *correct* kernels to *effective* ones, we must make rewards *bottleneck-aware*.

**Profiling-based Rewards (PR)**  A key symptom of this misalignment is the tendency for models to optimize trivial sub-operations (e.g., replacing a simple summation operation) without affecting the dominant bottlenecks in the computation. As demonstrated in the case study of lazy optimization versus better fusion (Figure 9), in the lazy optimization case, the model-generated kernel accounted for only $0.014\%$ of the total CUDA execution time, indicating that the kernel optimization did not affect the main bottlenecks. In contrast, with better fusion, the model generated kernels that covered $86.15\%$ of the total CUDA runtime, resulting in better and more meaningful speedup.

To formalize this intuition, we leverage the profiling toolkit in KERNELGYM to isolate the runtime contribution of the generated kernels ($T_{\text{generated}}$) from the overall CUDA execution time ($T_{\text{total}}$). We define the profiling ratio as:

$$\text{PR}_{i,t} = \frac{T_{\text{generated}}}{T_{\text{total}}}. \qquad (7)$$

Intuitively, $\text{PR}_{i,t}$ assigns higher credit when the candidate optimizes kernels that dominate the end-to-end runtime. We then augment the per-turn reward with this signal (applied only to correct kernels):

$$R_{i,t} = C(y_{i,t}) + C(y_{i,t}) \cdot \text{speedup}_{i,t} + C(y_{i,t}) \cdot \text{PR}_{i,t}. \qquad (8)$$

This encourages the model to focus on kernel optimizations that contribute significantly to performance, explicitly driving learning toward optimizations with larger real speedup. Besides, since $PR_{i,t}$ is bounded in $[0, 1]$, the speedup term naturally dominates, preventing the model from maximizing coverage via inefficient code.

**Profiling-based Rejection Sampling (PRS)**  Even with bottleneck-aware rewards, the exploration process can still be dominated by a high volume of low-impact ("lazy") samples. To further filter the training distribution, we introduce *profiling-based rejection sampling* (PRS). For each sample $(i, t)$, we retain it with probability:

$$p_{i,t} = \text{clip}\left(\frac{\text{PR}_{i,t} - \tau}{s}, 0, 1\right), \qquad (9)$$

where $\tau$ is a cutoff threshold and $s$ controls the softness of the filter. In our experiments, we fix $\tau = 0.3$ and $s = 0.1$. We ablate the design choice of PRS in Appendix D.

### 5.3. Empirical results

Figure 4 confirms this staged diagnosis. MRS improves training stability but does not, by itself, raise the Fast@1.2 ceiling. Furthermore, adding PR and PRS substantially lifts Fast@1.2. And the stability is even further improved by PR and PRS as shown in Figure 6.

## 6. Experiments

### 6.1. Setup

We evaluate on KernelBench (Ouyang et al., 2025) across all three levels. We follow the official Torch backend in Kernelbench and their implementations of correctness and speedup measurement. We furhter conduct hacking checks when we evaluate kernels. Therefore, our evaluations are stricter than the original Kernelbench. We follow the standard metrics in Kernelbench Fast@p, $p = [1, 1.2, 1.5, 2]$, where it is the ratio of samples are both correct and achieve $\times p$ speedup. We experiment with Qwen3-8B-Base and Qwen-14B-Base. We sample each question for 8 samples with 3 max turns. We set the max tokens as 32768 and the max generated tokens per turn as 8192. To keep a fair comparison, we report the results in the 3rd turn for all our models and most baselines since they always achieve the best performance in the 3rd turn. And for those who achieve the best performance before the 3rd turn, we report the results from previous turns.

We first perform cold-start supervised fine-tuning on our collected 8K 5-turn trajectories, using a learning rate of $1 \times 10^{-6}$, batch size 256, for 4 epochs. After cold-start training, we run multi-turn RL on the RL queries from cudaLLM (CudaLLM Team, 2025). The queries used for both SFT and RL cover basic PyTorch operators, Trans-

*Table 1.* Performance across levels under different Fast thresholds. As we treat samples with reward hacking as incorrect, our evaluation is more strict than the original Kernelbench. *The Cold-Start-8B refers to Qwen3-8B-Base after training with our cold-start data. We contain our DR. KERNEL-14B with sequential test-time scaling (STTS) using context management **??** as reference. DR. KERNEL-14B-STTS[†] reports the results from selecting the best turn across all history turns.

| Model | LEVEL1 | | | | LEVEL2 | | | | LEVEL3 | | | |
|---|---|---|---|---|---|---|---|---|---|---|---|---|
| | $Fast_1$ | $Fast_{1.2}$ | $Fast_{1.5}$ | $Fast_2$ | $Fast_1$ | $Fast_{1.2}$ | $Fast_{1.5}$ | $Fast_2$ | $Fast_1$ | $Fast_{1.2}$ | $Fast_{1.5}$ | $Fast_2$ |
| GPT-5 | 19.5 | 16.5 | 12.5 | 11.0 | 46.7 | 28.6 | 13.1 | 3.0 | 21.0 | 12.0 | 4.0 | 2.0 |
| Claude-4.5-Sonnet | 15.5 | 13.5 | 11.0 | 8.5 | 50.0 | 26.7 | 9.2 | 1.8 | 21.0 | 11.0 | 5.0 | 4.0 |
| Deepseek-V3.2-Thinking | 7.5 | 5.5 | 4.5 | 4.0 | 11.0 | 6.5 | 2.5 | 0.5 | 2.0 | 1.0 | 0.0 | 0.0 |
| GLM-4.7 | 19.4 | 17.2 | 13.1 | 10.4 | 30.0 | 20.5 | 8.5 | 3.5 | 5.0 | 2.0 | 2.0 | 2.0 |
| Qwen3-8B | 5.8 | 4.8 | 4.1 | 3.4 | 13.0 | 5.6 | 2 | 1.1 | 5.7 | 0.2 | 0.0 | 0.0 |
| Qwen3-32B | 6.1 | 4.9 | 4.3 | 4 | 14.0 | 9.4 | 2.4 | 0.2 | 3.5 | 0.0 | 0.0 | 0.0 |
| Qwen3-Coder-A30BA3 | 6.0 | 5.2 | 5.1 | 3.8 | 12.6 | 5.0 | 1.5 | 0.3 | 7.0 | 1.0 | 0.0 | 0.0 |
| AutoTriton | 4.5 | 3.6 | 2.8 | 2.1 | 30.6 | 9.2 | 2.6 | 0.5 | 7.5 | 0.0 | 0.0 | 0.0 |
| Cold-Start-8B* | 7.5 | 6.6 | 5.0 | 4.3 | 8.8 | 5.6 | 1.8 | 0.4 | 0.5 | 0.0 | 0.0 | 0.0 |
| DR. KERNEL-8B | 15.9 | 12.8 | 10.9 | 8.4 | 46.0 | 20.0 | 5.0 | 1.5 | **10.8** | 1.0 | 0.0 | 0.0 |
| DR. KERNEL-14B | **20.3** | **16.9** | **13.2** | **11.6** | **49.2** | **25.6** | **7.4** | **2.1** | 8.8 | **1.2** | 0.2 | 0.0 |
| DR. KERNEL-14B-STTS | 24.1 | 18.8 | 15.3 | 12.8 | 59.8 | 31.6 | 9.6 | 3.0 | 17.1 | 3.0 | 0.2 | 0.0 |
| DR. KERNEL-14B-STTS[†] | 39.3 | 25.1 | 20.4 | 17.6 | 80.9 | 47.8 | 23.6 | 12.5 | 29.8 | 7.3 | 0.5 | 0.0 |

former components, more complex compositions, and LLM-generated tasks. For RL, we use a learning rate of $1 \times 10^{-6}$, train for 300 rollout steps, sample 16 rollouts per prompt with max turns to 3 and a rollout batch size of 16. We implement the training pipeline with asynchronous inference.

## 6.2. Results

**Main Results.** Table 1 summarizes performance across KernelBench levels under different Fast thresholds. We compare DR. KERNEL against AutoTriton (Li et al., 2025), open-source models with strong coding/reasoning abilities, and proprietary models. AutoTriton is a very relevant baseline since it is released, Triton-based, and reports Fast@p.

Overall, DR. KERNEL achieves the strongest performance among open-source baselines and is competitive with frontier models on Level 1 and Level 2. In particular, DR. KERNEL-14B attains high Fast@1.2 on both Level 1 and Level 2, indicating that it improves not only *any* speedup (Fast@1) but also the stricter and meaningful speedup. This contrasts with prior approaches such as AutoTriton, which achieves a strong Fast@1 on Level 2 but delivers substantially smaller gains under stricter thresholds (e.g., Fast@1.2).

Comparing against our cold-start model, DR. KERNEL shows that multi-turn RL contributes materially to performance gains, especially on the stricter metrics (e.g., Fast@1.2 improves from $5.6 \rightarrow 20.0$ on Level 2). Finally, while DR. KERNEL improves Level 3 Fast@1 relative to open-source baselines, performance at stricter thresholds on Level 3 remains limited, suggesting that further scaling of training data and model capacity is likely required to close the gap to frontier models on the hardest subset.

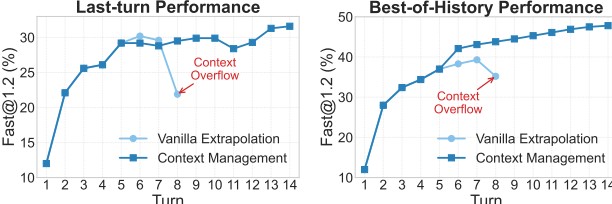

*Figure 5.* Test-time scaling with DR. KERNEL-14B. We report Fast@1.2 for the **last turn** (left) and **best-of-history** (right). Vanilla extrapolation increases the number of turns by appending all previous turns to the prompt. Context management stores the full history externally, but only includes the top-$w$ turns (by reward, $w=4$) in the prompt to fit context length.

**Analysis** We conduct a comprehensive analysis that includes reward hacking ratio (Appendix C) and case studies (Appendix E.2). Please refer to these sections for details.

## 6.3. Test-time Scaling for Kernel Generation

We study sequential test-time scaling (STTS) by increasing the number of multi-turn refinement steps at inference time. We use DR. KERNEL-14B with a maximum context length of 32,768 tokens and evaluate two strategies: **vanilla extrapolation** and **context management**. We report two metrics: (i) Last-turn Fast@1.2, i.e., the Fast@1.2 achieved by the final generated kernel at turn $T$; and (ii) Best-of-history Fast@1.2, i.e., the best Fast@1.2 obtained among turns $\{1, \ldots, T\}$. With STTS, our model even outperforms GPT-5/Claude-4.5-Sonnet in Kernelbench level-2 subset.

**Vanilla extrapolation** We directly extrapolate the number of refinement turns beyond training (trained with up to 3 turns) by appending the entire interaction history to the prompt at each turn. Figure 5 shows that increasing turns initially

*Table 2.* Fast performance across levels under different thresholds (evaluated under `torch.compile`). Since `torch.compile` provides a strong optimized baseline, Fast@1 remains meaningful (unlike eager mode, where trivial "lazy" optimizations can inflate Fast@1).

| Model | LEVEL1 | | | | LEVEL2 | | | | LEVEL3 | | | |
|---|---|---|---|---|---|---|---|---|---|---|---|---|
| | $Fast_1$ | $Fast_{1.2}$ | $Fast_{1.5}$ | $Fast_2$ | $Fast_1$ | $Fast_{1.2}$ | $Fast_{1.5}$ | $Fast_2$ | $Fast_1$ | $Fast_{1.2}$ | $Fast_{1.5}$ | $Fast_2$ |
| GPT-5 | 18.6 | 8.0 | 6.5 | 5.5 | 22.1 | 3.6 | 1.5 | 1.0 | 14.0 | 4.0 | 3.0 | 1.0 |
| Claude-4.5-Sonnet | 10.0 | 2.2 | 2.0 | 1.8 | 20.5 | 3.0 | 0.0 | 0.0 | 12.0 | 3.5 | 0.5 | 0.0 |
| DR. KERNEL-8B | 16 | 3.0 | 1.5 | 8.4 | 20.6 | 0.8 | 0.0 | 0.0 | 7.2 | 2.3 | 0.0 | 0.0 |
| DR. KERNEL-14B | 17.8 | 5 | 3.3 | 2.5 | 23.5 | 1.9 | 0.0 | 0.0 | 9.2 | 3 | 0.0 | 0.0 |

improves both last-turn and best-of-history performance. However, as $T$ grows, prompt length scales linearly and may approach the context limit, which can degrade performance.

**Context management** To scale $T$ without unbounded prompt growth, we store all turns in an external memory and maintain a fixed in-context window. Concretely, at each turn we select the top-$w$ turns with the highest rewards from the accumulated history and only append these selected turns as the prompt history for generating the next turn (we use $w=4$). As shown in Figure 5, context management yields consistently stronger best-of-history performance and continues to improve as turns scale. Last-turn performance can be slightly lower at small $T$, since vanilla extrapolation can condition on the full history, but as $T$ increases, context management becomes strictly more reliable and surpasses the best performance achievable by vanilla extrapolation.

### 6.4. Results on `torch.compile`

`torch.compile` is an advanced PyTorch feature that captures PyTorch programs into a compiled computation graph and optimizes execution via operator fusion, code generation, and scheduling, among other compiler passes. While most prior work evaluates model-generated kernels only under Torch eager execution, we further validate both our models and frontier models under `torch.compile`, providing a substantially stronger and more practical assessment of speedups.

As shown in Table 2, DR. KERNEL remains effective under the more challenging `torch.compile` setting and stays competitive with frontier models across the three levels. Because `torch.compile` already applies compiler optimizations, the headroom for additional gains is smaller than in eager mode; consequently, the absolute Fast@p numbers are generally lower for all models, including both ours and frontier models. Importantly, Fast@1 under `torch.compile` is also a stricter target: trivial "lazy" changes that may yield marginal improvements in eager execution typically do not surpass the optimized compiled baseline. These results suggest that our training methods generalize beyond eager-mode artifacts, and further scaling

of data and model capacity is a promising direction to obtain larger improvements even on top of `torch.compile`.

## 7. Conclusion

We investigate RL for Triton kernel generation and identify the challenges of *reward hacking* and *lazy optimization*, which are common in prior works. To address these, we develop the KERNELGYM environment with hacking checks and profiling tools, propose unbiased multi-turn RL methods incorporating mismatch correction, profiling-based rewards/rejection sampling, and explore sequential test-time scaling. Our approach enhances meaningful speedup, advancing RL training for kernel generation.

## Impact Statement

This paper presents work whose goal is to advance LLMs for kernel generations. There are many potential societal consequences of our work, none which we feel must be specifically highlighted here.

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

## A. Derivation: Self-Inclusion Causes a Scaled Gradient in GRPO

**Setup** For a given prompt question, fix a turn group $\mathcal{G}_t$ with $|\mathcal{G}_t| = N$. For each rollout $i \in \mathcal{G}_t$, let $s_{i,t} \triangleq (h_{i,:t-1}, x_{i,t})$ and $y_{i,t} \sim \pi_\theta(\cdot \mid s_{i,t})$, where $h_{i,:t-1}$ is all turn history before $t$, $x_{i,t}$ is the prompt with environmental feedback in turn $t$. Let $G_{i,t}$ be the reward-to-go return at turn $t$ and define the in-group mean $\bar{G}_t = \frac{1}{N} \sum_{j \in \mathcal{G}_t} G_{j,t}$. GRPO mean-centering uses $A_{i,t}^{\text{GRPO}} = G_{i,t} - \bar{G}_t$.

For any random variable $Z$ that does *not* depend on the sampled action $y_{i,t}$ (conditional on $s_{i,t}$),

$$\mathbb{E}_{y_{i,t} \sim \pi_\theta(\cdot \mid s_{i,t})}\big[\nabla_\theta \log \pi_\theta(y_{i,t} \mid s_{i,t}) \cdot Z\big] = Z \cdot \mathbb{E}_{y_{i,t}}\big[\nabla_\theta \log \pi_\theta(y_{i,t} \mid s_{i,t})\big] = 0, \tag{10}$$

where we use $\mathbb{E}_y[\nabla_\theta \log \pi_\theta(y \mid s)] = \nabla_\theta \int \pi_\theta(y \mid s)\, dy = \nabla_\theta 1 = 0$.

**Policy Gradient in GRPO** Consider the per-turn GRPO estimator within this group:

$$\hat{g} = \frac{1}{N} \sum_{i \in \mathcal{G}_t} \nabla_\theta \log \pi_\theta(y_{i,t} \mid s_{i,t}) \left(G_{i,t} - \bar{G}_t\right). \tag{11}$$

Taking expectation and expanding the baseline term,

$$\mathbb{E}[\hat{g}] = \frac{1}{N} \sum_i \mathbb{E}[\nabla_\theta \log \pi_\theta(y_{i,t} \mid s_{i,t})\, G_{i,t}] - \frac{1}{N} \sum_i \mathbb{E}\big[\nabla_\theta \log \pi_\theta(y_{i,t} \mid s_{i,t})\, \bar{G}_t\big]$$

$$= \frac{1}{N} \sum_i \mathbb{E}[\nabla_\theta \log \pi_\theta(y_{i,t} \mid s_{i,t})\, G_{i,t}] - \frac{1}{N} \sum_i \mathbb{E}\left[\nabla_\theta \log \pi_\theta(y_{i,t} \mid s_{i,t}) \frac{1}{N} \sum_j G_{j,t}\right]. \tag{12}$$

For $j \neq i$, $G_{j,t}$ is independent of $y_{i,t}$ under independent rollouts, hence we can apply the score-function identity in Eq. (10) to obtain

$$\mathbb{E}[\nabla_\theta \log \pi_\theta(y_{i,t} \mid s_{i,t})\, G_{j,t}] = 0 \qquad (j \neq i).$$

Therefore only the $j = i$ term remains:

$$\mathbb{E}[\hat{g}] = \frac{1}{N} \sum_i \mathbb{E}[\nabla_\theta \log \pi_\theta(y_{i,t} \mid s_{i,t})\, G_{i,t}] - \frac{1}{N} \sum_i \mathbb{E}\left[\nabla_\theta \log \pi_\theta(y_{i,t} \mid s_{i,t}) \frac{1}{N} G_{i,t}\right]$$

$$= \left(1 - \frac{1}{N}\right) \cdot \frac{1}{N} \sum_i \mathbb{E}[\nabla_\theta \log \pi_\theta(y_{i,t} \mid s_{i,t})\, G_{i,t}]. \tag{13}$$

The last term corresponds to the unbiased REINFORCE gradient for this prompt–turn group (up to the outer averaging convention), hence the GRPO mean baseline induces a shrinkage factor $(1 - \frac{1}{N})$ due to self-inclusion.

**Leave-one-out removes self-inclusion.** Using the leave-one-out baseline $\bar{G}_t^{(-i)} = \frac{1}{N-1} \sum_{j \neq i} G_{j,t}$, the baseline term no longer contains $G_{i,t}$ and is independent of $y_{i,t}$ under independent rollouts. By Eq. (10), the expected contribution of the baseline term becomes zero, yielding an unbiased estimator.

## B. Training Dynamics

In this section, we analyze the training dynamics. As shown in Figure 6, multi-turn RL for kernel generation exhibits significant instability, characterized by elevated entropy, perplexity, and gradient norms, even when using unbiased advantage estimation. Incorporating mismatch correction (i.e., Mismatch Rejection Sampling) effectively stabilizes the training process. Furthermore, the introduction of PR and PRS provides additional stability, leading to smoother training.

## C. Hacking Ratio

We analyze the changes in the hacking ratio during training for DR. KERNEL-14B on the Kernelbench level-2 subset. With the hacking check in KERNELGYM, the hacking ratio steadily decreases from approximately 20% at the start to around 3%. We also examine the hacking ratio on the Kernelbench level-1 subset; compared to AutoTriton, which exhibits a hacking ratio of around 10%, DR. KERNEL-14B experiences hacking in only 1.7% of cases.

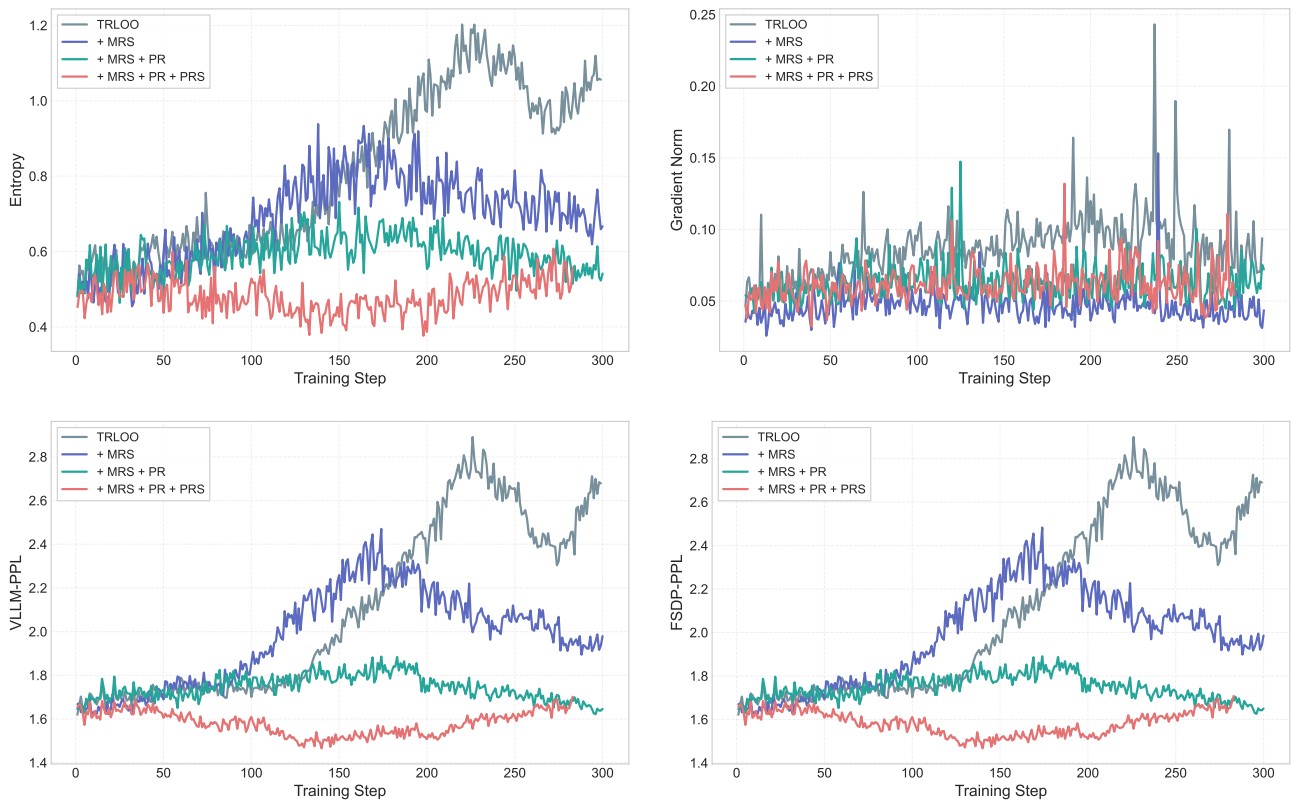

*Figure 6.* The training dynamics of TRLOO, TRLOO + Mismatch Rejection Sampling (MRS), TRLOO + MRS + Profiling-based Reward (PR) and TRLOO + MRS + PR + Profiling-based Rejection Sampling (PRS). We analyze the training dynamics via the lens of entropy, gradient norm, VLLM-PPL, and FSDP-PPL, which are also monitored by Liu et al. (2025).

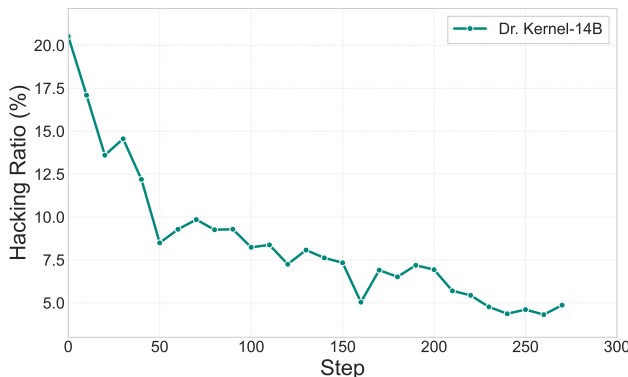

*Figure 7.* The hacking ratio of DR. KERNEL-14B. With the hacking check in KERNELGYM, the hacking ratio decreases from approximately 20% at the start to only around 3% in the Kernelbench level-2 subset.

## D. Ablations of PRS

We perform an ablation study to evaluate the effect of the softness sampling design choice in PRS. Our default setting for DR. KERNEL is the combination of TRLOO, MRS, PR, and PRS, and we compare it to a variant, DR. KERNEL$_{\text{w/o s in PRS}}$, where kernels with $PR \geq \tau$ are kept directly, and kernels with $PR < \tau$ are discarded outright. In contrast, in the variant with softness, kernels with $PR \in [0.3, 0.4)$ are probabilistically retained, based on $PR$, $\tau$, and $s$. In this ablation, we set $\tau = 0.3$ as the fixed configuration.

As shown in Figure 8, DR. KERNEL outperforms the variant `w/o s in PRS`. The variant still performs better and shows improved stability compared to the baseline `w/o PR & PRS`. This ablation demonstrates that the softness sampling mechanism enhances the robustness of PRS, helping it balance correctness and meaningful speedup by selectively retaining relatively low-quality samples that might otherwise be discarded. This ability to retain such samples contributes to a more effective exploration-exploitation trade-off and facilitates more stable kernel generation over time.

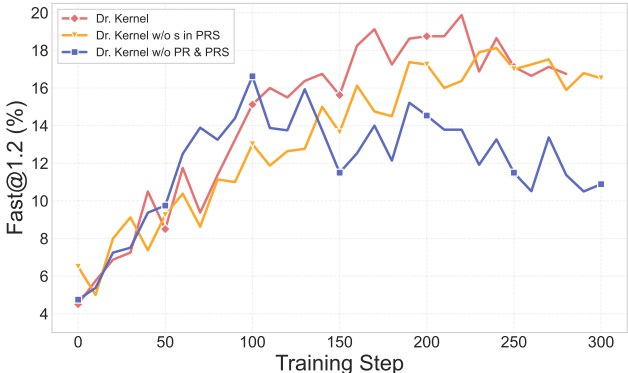

*Figure 8.* Comparison of DR. KERNEL with and without softness in PRS. DR. KERNEL outperforms the variant `w/o s in PRS`, while both variants show better stability compared to the baseline `w/o PR & PRS`.

## E. Cases Studies

### E.1. Lazy Optimization vs. Better Fusion

We show the cases of profiling feedback from lazy optimization and better fusion cases. As shown in Figure 9, in the lazy optimization case, where only a trivial summation operation is replaced, the model-generated kernel accounts for only $0.014\%$ of the total CUDA execution time. In contrast, with better fusion, the model generates more meaningful kernels, achieving significantly better speedup and increasing the CUDA runtime fraction to $86.15\%$ of the total runtime.

### E.2. Trajectories from DR. KERNEL

We conduct qualitative studies on DR. KERNEL-14B using multi-turn inference across three turns. As shown in Figure 10, in the first turn, DR. KERNEL identifies the LayerNorm operation and generates a kernel by fusing different operations into a single Triton kernel. After receiving feedback from KERNELGYM, it recognizes that certain configurations, such as block size, number of wraps, and stages, are under-explored, and applies "autoconfig" to select better configurations. By the third turn, DR. KERNEL identifies the optimal configuration based on the running hardware, further improving the performance by adjusting the configuration in "autoconfig." This case demonstrates that after RL training, DR. KERNEL-14B handles basic kernel writing and also adapts effectively to environment feedback, showcasing its ability to improve over time.

### E.3. Case for Better Fusion

We further study the case where the lazy optimization issue is alleviated, as shown in the profiling summary in Figure 9 (right). In contrast to the lazy optimization case, the example generated by DR. KERNEL-8B addresses lazy optimization by converting most operations into Triton kernels, which account for $86.15\%$ of the total CUDA runtime. However, we note that a convolution operation remains in the implementation. During training, we observed that the model attempted to implement this operation with Triton. Despite Triton's potential for kernel optimization, convolution operations are difficult

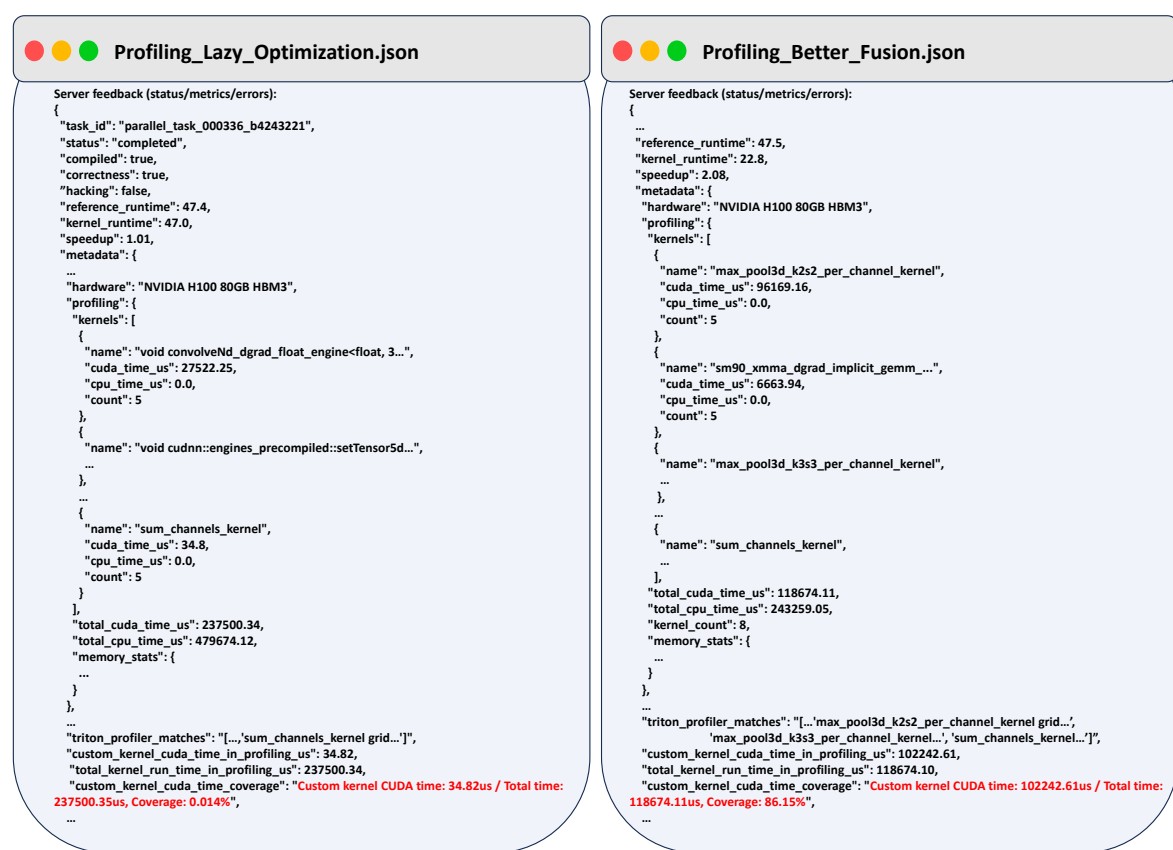

*Figure 9.* Profiling feedback for cases with lazy optimization and better fusion. We omit some profiling items for brevity. The original code for `Profiling_Lazy_Optimization.json` is shown in Figure 1 (right), and the original code for `Profiling_Better_Fusion.json` can be found in Appendix E.3. In the lazy optimization case, where only a trivial summation operation is replaced, the model-generated kernel accounts for only $0.014\%$ (i.e $PR = 0.00014$) of the total CUDA execution time. In contrast, with better fusion, the model generates more meaningful kernels, achieving significantly better speedup and increasing the CUDA runtime fraction to $86.15\%$ i.e $PR = 0.8615$ of the total runtime.

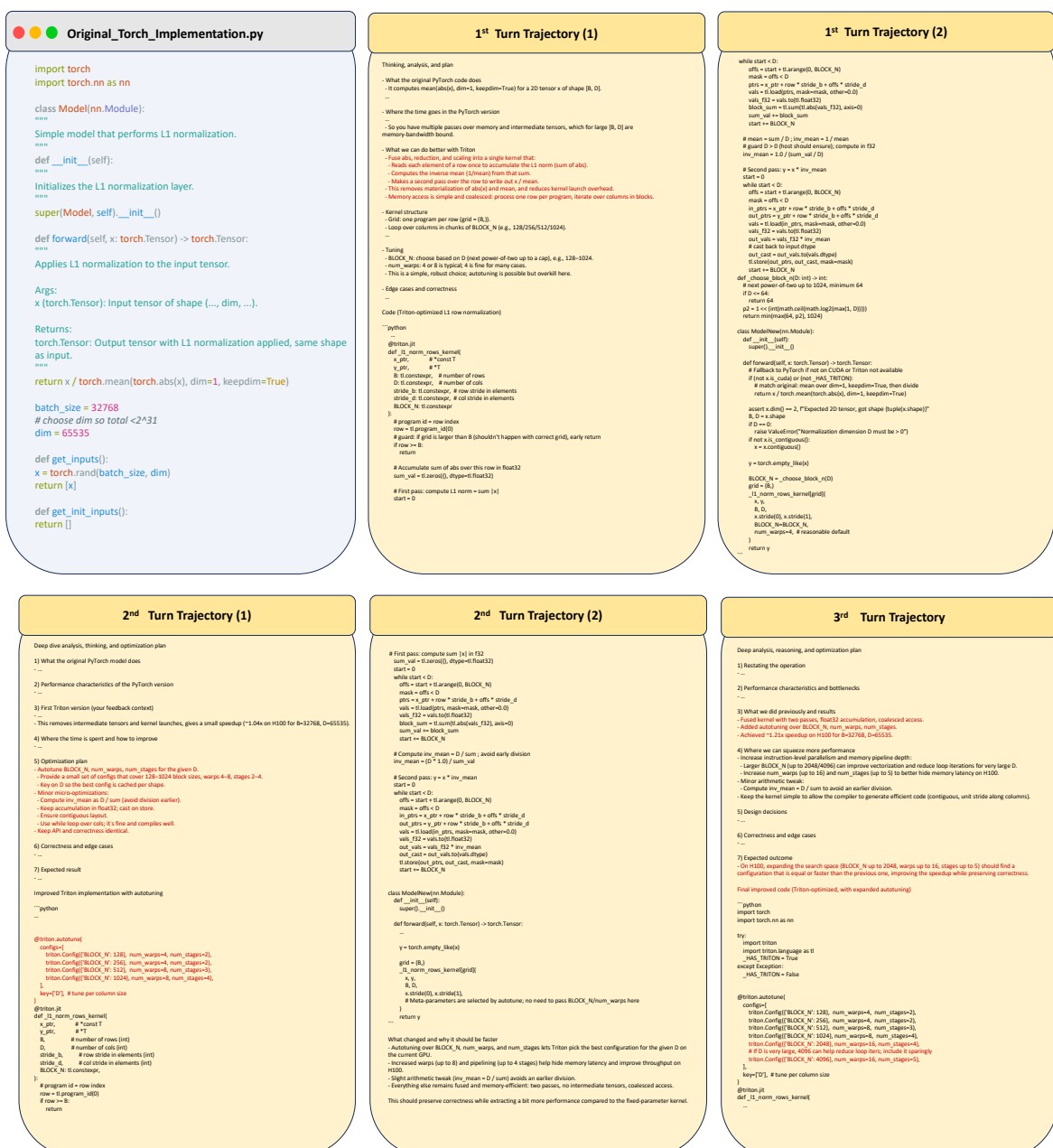

*Figure 10.* The case study of DR. KERNEL-14B in LayerNorm operation from Kernelbench Level-1 subset. The speedup of the three turns are 1.04, 1.21, and 1.45, respectively. In this case, DR. KERNEL-14B effectively handles basic kernel writing and adapts to environmental feedback over multiple turns.

to implement effectively, as they are highly optimized by libraries such as cuDNN. Consequently, our small-sized models with limited data struggle to implement this operation as effectively as cuDNN, and instead, the model focuses on fusing other kernels. This behavior highlights one of the root causes of lazy optimization and emphasizes the importance of a well-established optimization objective in RL-based kernel generation, such as a bottleneck-aware profiling-based reward.

```python
class ModelNew(nn.Module):
    def __init__(self, ...):
        ...

    def forward(self, x):
        # ConvTranspose3d using PyTorch/cuDNN
        x = self.conv_transpose(x) # x shape: [N, out_channels, D, H, W]
        ...
        if use_triton:
            # First pooling: kernel=2, stride=2, padding=0
            D_out1 = (D - 2) // 2 + 1
            H_out1 = (H - 2) // 2 + 1
            W_out1 = (W - 2) // 2 + 1

            x_pooled1 = torch.empty((N, C, D_out1, H_out1, W_out1), dtype=x.dtype, device=x.device)

    grid1 = (N, C, D_out1 * H_out1 * W_out1)
    max_pool3d_k2s2_per_channel_kernel[grid1](
    x, x_pooled1,
    N, C, D, H, W,
    D_out1, H_out1, W_out1,
    x.stride(0), x.stride(1), x.stride(2), x.stride(3), x.stride(4),
    x_pooled1.stride(0), x_pooled1.stride(1), x_pooled1.stride(2), x_pooled1.stride(3),
    x_pooled1.stride(4),
    num_warps=4, num_stages=2
    )

    # Second pooling: kernel=3, stride=3, padding=0
    D_out2 = (D_out1 - 3) // 3 + 1
    H_out2 = (H_out1 - 3) // 3 + 1
    W_out2 = (W_out1 - 3) // 3 + 1

    x_pooled2 = torch.empty((N, C, D_out2, H_out2, W_out2), dtype=x.dtype, device=x.device)

    grid2 = (N, C, D_out2 * H_out2 * W_out2)
    max_pool3d_k3s3_per_channel_kernel[grid2](...)

    ...
    grid_sum = (N * D_out2 * H_out2 * W_out2,)
    sum_channels_kernel[grid_sum](
    x_pooled2, y,
    N, C, D_out2, H_out2, W_out2,
    x_pooled2.stride(0), x_pooled2.stride(1), x_pooled2.stride(2), x_pooled2.stride(3),
    x_pooled2.stride(4),
    y.stride(0), y.stride(2), y.stride(3), y.stride(4),
    num_warps=4, num_stages=2
    )

    return y
```

*Figure 11.* The case study of better fusion from Dr. Kernel-8B, which corresponds to the profiling feedback in Figure 9 (right).

## F. Prompt Template

### F.1. Prompt Template for Cold-Start Data Distillation

We show the template for cold-start data distillaion in Figure 12.

### F.2. Prompt Template for SFT and RL

We show the template for both SFT and RL in Figure 13.

## 1st Turn Prompt Template for Training

You are looking at this PyTorch code and thinking it could be optimized with Triton.

Here's the PyTorch code:

```python
{reference_code}
```

You need to create a Triton version with the entry point called `{entry_point}New`.

Please firstly analyze this code and think hard how you can optimize it.

**Please output and show your thinking, plan, analysis etc., before your coding, which should be as more as possible.**

## Prompt Template After 1st Turn

Server feedback from the evaluation environment for your last implementation: {kernelgym_feedbacks}

Based on the above server feedback, please improve the implementation:
- If there are errors/crashes/illegal memory access: identify the root cause and fix it; prevent recurrence.
- If there is no speedup or performance regresses: optimize the bottlenecks to achieve a clear speedup.
- If there is already a speedup: further improve performance without degrading correctness.
- Please output your thinking, plan, analysis, and the final code.

*Figure 12.* The prompt template for cold-start data distillation.

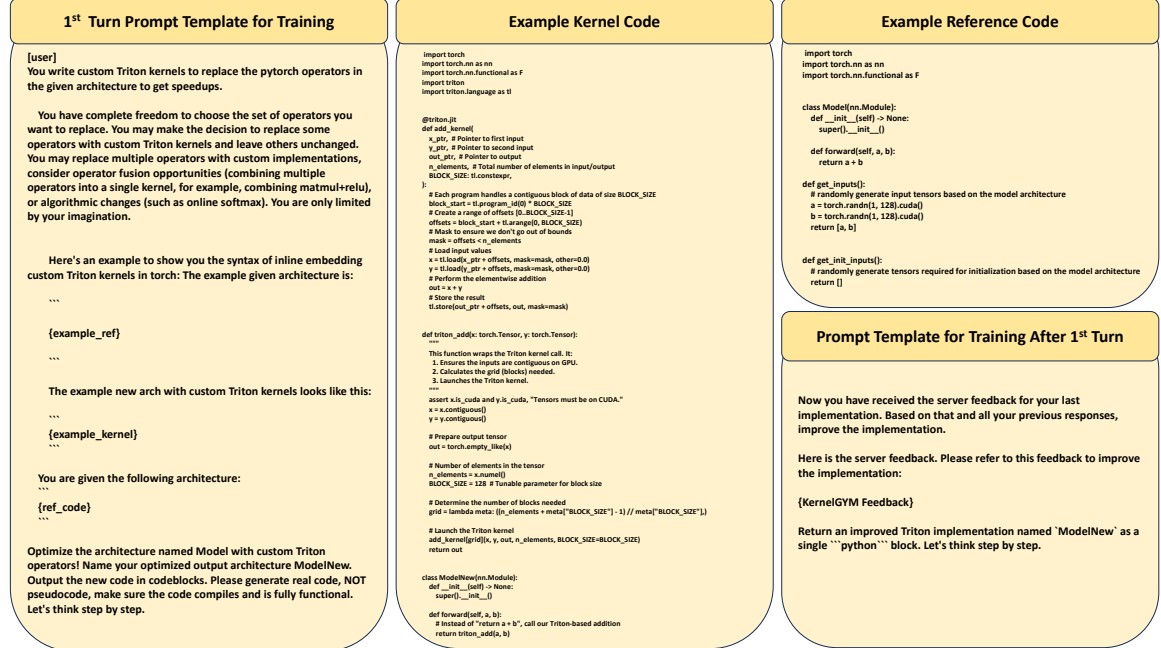

*Figure 13.* The prompt template for both SFT and RL. The task instruction, example code and reference code are shown in the first-turn prompt. And for the later turns, prompt asks model to refine the kernel implementaion based on all hisotory information and KERNELGYM feedbacks.

