# OpenReview forum: "Dr. Kernel: Reinforcement Learning Done Right for Triton Kernel Generations"
_ICML.cc/2026/Conference — ICML 2026 regular_

### Official Review · Reviewer_5dLi · 2026-02-26

**Soundness:** 3
**Presentation:** 3
**Significance:** 4
**Originality:** 3
**Overall Recommendation:** 4
**Confidence:** 4

**Summary:**

This paper has 2 major contributions: First, a robust reinforcement learning environment, KERNELGYM, for Triton-based automated GPU kernel generation. Second, well-designed RL polices in this domain with comprehensive experiments and evaluation. The proposed KERNELGYM  implements a scalable distributed RL system, enabling high-quality multi turn roll out data collection. For GPU kernel generation, proposed RL policies not only covers correctness and performance, but also considers "lazy optimization" problem in automated LLM-based GPU kernel generation. Experiment results show that the trained model DR.Kernel-14B can achieve comparable performance with frontier models on parts of tasks on KernelBench.

**Compliance With Llm Reviewing Policy:**

Affirmed.

**Final Justification:**

Authors answered most of my confusion in rebuttal, however, the test setting is still very limited and the effectiveness or their RL method seems not promising. For example, the test set is limited to level2 on KernelBench and the reason they claimed is that "level 2 fits better to current LLM's ability"(I used to think that is their training set but they claim that it is only test set in rebuttal). Actually many of related works have done a lot of experiments on full KernelBench(some of them even have been cited by this paper), which means this claim and test setting is not reasonable. Further, the ability of their RL model showed is very limited. Considering the high cost they need for their proposed RL pipeline(has been proved in other reviewers' questions), it is really questionable if their RL can really achieve good results eventually.

According to above all, I choose to maintain my final score for this paper.

[update]

Authors further clarify some questions. I decide to give 4 as my score.

**Key Questions For Authors:**

I have following questions for authors, and I'm willing to enhance my score if they can address my questions.

1. According to my experience on KernelBench and other related benchmarks, there are some bugs and unreasonable settings on them, which can lead to some misleading results(for example LLM can get 5-10x speedup without hacking). For example, the task 80 on level 2 of KernelBench, where no matter what the input is the output will be just all 0 tensors. Have you noticed this in your experiments and address them?

2.  What information and metrics will be included in your profiling during training and inference? Only the runtime of current kernels?

3. In proposed bottleneck-aware reward design, you decide identify if bottleneck is addressed through computing T_generated/T_total. Is it possible that although generated kernel cover most of program runtime but actually it is still not real bottleneck? Because the generated kernel's performance can be very bad, leading new generated kernel becomes current bottleneck. But actually that kernel may not the real bottleneck in original reference PyTorch codes.  Current LLMs can fail on easy tasks.

4. With so complex design and high-cost training, model still performed bad on Level3, can you give some reasons and analysis on this? What is the challenges behind level3?

5. Why only training on Level2? Although you explained this with "level 2 fits better to current LLM's ability", this claim is not solid. And is there any overlapping between your training set and test set?

**Limitations:**

yes

**Strengths And Weaknesses:**

Strengths:
1. Good soundness and presentation, writing and presentation are clear and solid.
2. Identify some critical problems in automated GPU kernel generation with LLM and proposed practical solutions.
3. Methodology design is original and novel.

Weakness:
All proposed problems and solutions seem promising, however, eventual performance seems not that good. The robustness of results showed in paper is also uncertain. Testing only cover sub parts of KernelBench.

---

> ### Author Rebuttal · Authors · 2026-03-31
>
> Thanks for your review and suggestions.
>
> # W1
> ```
> ...eventual performance seems not that good...The robustness of results showed in paper is also uncertain. Training only cover sub parts of KernelBench.
> ```
> Our models achieve substantial gains over their baselines and even outperform much larger models such as GLM-4.7.
>
> We conducted three independent runs for TRLOO+MRS, TRLOO+MRS+PR, and TRLOO+MRS+PR+PRS on both 8B and 14B models. The Fast@1.2 results are provided in **Reviewer 2JS5, W1**. Improvements are consistent across two sizes, and the low standard deviations (<0.5 for 8B, <0.3 for 14B) support the reliability of our RL results.
>
> We also emphasize we do not use any data from KernelBench for training. As described in Sections 4.1 and 6.1, both SFT and RL use queries from open-source datasets, with no overlap with the KernelBench test set.
>
> # Q1
> ```
> According to my experience on KernelBench..., there are some bugs and unreasonable settings on them, which can lead to some misleading results... For example, the task 80 on level 2 of KernelBench, where no matter what the input is the output will be just all 0 tensors. Have you noticed this in your experiments and address them?
> ```
> We agree that some early KernelBench contains problematic settings. In our experiments, we use an officially updated version of KernelBench, where several known issues have already been fixed.
>
> To further enhance the evaluation, we manually identified 7 potentially problematic Level 2 tasks, also discussed in open issues [1], and removed them. These cases fall into two types: (1) algebraically simplifiable operations, where a model could obtain “free” speedup through mathematical shortcuts, (2) degenerate or buggy operations, such as redundant double pooling or always-zero outputs, including Task 80.
>
> The sanitized Level 2 results are shown below:
> |  |Original (100)|Sanitized (93)|
> |---|---:|---:|
> |FAST@1|49.2%|50.4%|
> |FAST@1.2|25.6%|26.1%|
> |FAST@1.5|7.4%|6.8%|
> |FAST@2|2.1%|2.0%|
> These results suggest that our model does not rely on such patterns to achieve speedup, and our main conclusions remain unchanged after sanitization.
>
> [1] www.github.com/BytedTsinghua-SIA/CUDA-Agent/issues/7
>
> # Q2
> ```
> What information and metrics will be included in your profiling during training and inference? Only the runtime of current kernels?
> ```
> For kernels that fail to execute, we provide a full error traceback pinpointing the failure line, which serves as critical feedback for self-correction.
>
> For successful kernels, we collect detailed performance data (Figure 9), including cudaEvent timings and memory/runtime stats from torch.profiler. We also compute secondary metrics such as relative speedup against the PyTorch baseline.
>
> # Q3
> ```
> In proposed bottleneck-aware reward design, you decide identify if bottleneck is addressed through computing T_generated/T_total. Is it possible that although generated kernel cover most of program runtime but actually it is still not real bottleneck? Because the generated kernel's performance can be very bad, leading new generated kernel becomes current bottleneck...
> ```
> Our method does not suffer from this issue because the speedup term dominates the reward. As shown in Eq. 8, the profiling ratio $PR_{i,t}$ is bounded in [0,1], while an inefficient generated kernel directly reduces the speedup term. Therefore, maximizing coverage with slow code cannot improve the overall reward; the model must generate kernels that actually run faster.
>
> # Q4
> ```
> With so complex design and high-cost training, model still performed bad on Level3, can you give some reasons and analysis on this? What is the challenges behind level3?
> ```
> Our training cost is standard for agentic RL, requiring only 3-4 H100 nodes for 1-2 days. Level 3 tasks involve highly complex full-model architectures such as MiniGPT. While our models do not yet outperform closed-source frontier models like GPT-5 or Claude on this subset, DR. KERNEL-8B and 14B still consistently outperform all other open-source baselines, including GLM-4.7.
>
> The gap on Level 3 is mainly due to model capacity (model size and pre-training data). For these highly complex tasks, 8B/14B models struggle to generate correct implementations. Since RL depends on high-quality successful rollouts, and such explorations are extremely sparse on Level 3, the RL process is bottlenecked by the base model’s limitations.
>
> # Q5
> ```
> Why only training on Level2? Although you explained this with "level 2 fits better to current LLM's ability", this claim is not solid. And is there any overlapping between your training set and test set?
> ```
> We do not train on Level 2; we use it only for validation during RL training. The reason is efficiency: full-benchmark validation during RL is time-consuming, while Level 2 already contains fused multi-operator kernels but is less complex than Level 3, making it a practical validation set.
>
> We also do not use any data from KernelBench for training, as clarified in W1.

---

> > ### Author Rebuttal · Reviewer_5dLi · 2026-04-01
> >
> > Authors answered all of my questions in rebuttal, I will give my final recommendation accordingly.

---

> > > ### Author Response · Authors · 2026-04-07
> > >
> > > Dear Reviewer 5dLi,
> > >
> > > Thank you again for your time. We appreciate your acknowledgment that our rebuttal fully resolved your concerns. However, we noticed that some points in the final justification appear to be based on a misunderstanding of our paper, and we would like to respectfully clarify.
> > >
> > > # Clarification 1: We evaluate on the full KernelBench (all 3 levels), not just Level 2
> > >
> > > ```
> > > the test set is limited to level2 on KernelBench
> > > ```
> > >
> > > This is not the case. **Table 1 in our paper reports results on all three levels of KernelBench** (Level 1, Level 2, and Level 3), which is the full benchmark. Level 2 is used only as a validation set during RL training for efficiency, as we clarified in Q5 of our rebuttal. Our evaluation covers the complete KernelBench, the same as other related works.
> > >
> > > # Clarification 2: RL improvements are significant across all levels and model sizes
> > >
> > > ```
> > > the effectiveness of their RL method seems not promising... the ability of their RL model showed is very limited
> > > ```
> > >
> > > We respectfully disagree. We reproduce key results from Table 1 below for convenience. RL training yields **consistent and significant improvements** across all three levels and both model sizes:
> > >
> > > | Model | Level 1 Fast@1 | Level 1 Fast@1.2 | Level 2 Fast@1 | Level 2 Fast@1.2 | Level 3 Fast@1 |
> > > |:---|---:|---:|---:|---:|---:|
> > > | GPT-5 | 19.5 | 16.5 | 46.7 | 28.6 | 21.0 |
> > > | Claude-4.5-Sonnet | 15.5 | 13.5 | 50.0 | 26.7 | 21.0 |
> > > | GLM-4.7 | 19.4 | 17.2 | 30.0 | 20.5 | 5.0 |
> > > | Qwen3-8B | 5.8 | 4.8 | 13.0 | 5.6 | 5.7 |
> > > | Cold-Start-8B | 7.5 | 6.6 | 8.8 | 5.6 | 0.5 |
> > > | **Dr. Kernel-8B** | **15.9** | **12.8** | **46.0** | **20.0** | **10.8** |
> > > | **Dr. Kernel-14B** | **20.3** | **16.9** | **49.2** | **25.6** | **8.8** |
> > > | Dr. Kernel-14B-STTS | 24.1 | 18.8 | 59.8 | 31.6 | 17.1 |
> > > | Dr. Kernel-14B-STTS† | 39.3 | 25.1 | 80.9 | 47.8 | 29.8 |
> > >
> > > Dr. Kernel-8B improves over Cold-Start-8B by **2×–5×** on Fast@1/Fast@1.2 across all levels. Dr. Kernel-14B achieves performance competitive with frontier models such as GPT-5 and Claude-4.5-Sonnet, despite being orders of magnitude smaller. With sequential test-time scaling (STTS), Dr. Kernel-14B-STTS further surpasses frontier models on all levels. STTS† (selecting the best turn across all history turns) pushes performance even further, reaching 80.9 Fast@1 on Level 2 and 29.8 Fast@1 on Level 3.
> > >
> > > Additionally, as we reported to Reviewer LX8g, Dr. Kernel-14B achieves **1.41× speedup** on the TriMul kernel (Triangle Multiplicative Update, a core operation in AlphaFold3), competitive with GPT-5 (1.31×) and Claude-4.5-Sonnet (1.39×), demonstrating strong cross-benchmark generalization beyond KernelBench.
> > >
> > > Reviewer 2JS5 noted that "Dr Kernel seems to **outperform even frontier LLMs** on relevant tasks while only having 14B parameters".
> > >
> > > # Clarification 3: RL training cost is standard
> > >
> > > ```
> > > Considering the high cost they need for their proposed RL pipeline
> > > ```
> > >
> > > As we clarified in our rebuttal to Reviewer 44jY (W2), RL training uses only 2 H100 nodes for training + 1–2 nodes for evaluation, completing in 1.5–2 days. This is comparable to other agentic RL work [1] and does not constitute a high cost.
> > >
> > > We hope these clarifications address the misunderstandings in the final justification. We sincerely welcome any further questions, and would appreciate your reconsideration in light of the above.
> > >
> > > [1] Xue et al., SimpleTIR: End-to-End Reinforcement Learning for Multi-Turn Tool-Integrated Reasoning, https://arxiv.org/abs/2509.02479

---

### Official Review · Reviewer_44jY · 2026-03-10

**Soundness:** 3
**Presentation:** 3
**Significance:** 3
**Originality:** 3
**Overall Recommendation:** 4
**Confidence:** 4

**Summary:**

The paper systematically explores using RL-based LLM for Triton GPU kernel generation. It introduces KernelGym，a robust kernel evaluation environment，supporting hacking check and profiling feedback. It introdices TRLOO (Turn-level REINFORCE Leave-One-Out) as advantage estimator. To address lazy optimization, it introduces profiling reward(PR)and ejection sampling(PRS). The Dr. Kernel-14B is comparable to Claude-4.5-Sonnet in KernelBench.

**Compliance With Llm Reviewing Policy:**

Affirmed.

**Final Justification:**

The authors successfully addressed all my concerns, and I also saw that they had meaningful discussion with other reviewers, resloving many important problems. I am happy to raise my score to 4.

**Key Questions For Authors:**

I think this paper is a good paper. However, the weaknesses need to be addressed.

Moreover,
1. Could you provide more cases about optimized kernels?

2. Could you report avg speedup in the results?

I am willing to update my scores, based on author's rebuttal

**Limitations:**

yes

**Strengths And Weaknesses:**

Strength:

1. The design of KernelGYM is very comprehensive and robust. It is highly demanded in current kernel generation research.

2. Profiling-based reward is novel and interesting. It solves an important problem in kernel related training

3. The ablation study is very comprehensive. It clearly demonstrates each conponent's contribution in the whole system.


Weaknesses:

1. The contribution of PRS is not very obvious. From the Figure 4, I think the results are almost the same, regardless with or w/o PRS.

2. It seems that the training of Dr. Kernel is highly expensive. I think author should add some details about GPU/Time consumption in RL.

3. How the Dr. Kernel do multi-turn generation is not very clear. Authors should add details. For example, How they organize context? / Do they use tools/profiling in inference?

---

> ### Author Rebuttal · Authors · 2026-03-31
>
> Thanks for your review and suggestions.
>
> # W1
>
> ```
> The contribution of PRS is not very obvious. From the Figure 4, I think the results are almost the same, regardless with or w/o PRS.
> ```
>
> Here to better assess the contribution of PRS, we conducted three independent runs for TRLOO+MRS, TRLOO+MRS+PR, and TRLOO+MRS+PR+PRS for both the 8B and 14B models.
>
> |Method|Run1|Run2|Run3|Mean|Std|
> |---|---|---|---|---|---|
> |TRLOO+MRS(8B)|17.50|17.80|17.00|17.43|0.40|
> |TRLOO+MRS+PR(8B)|18.60|18.00|18.25|18.28|0.30|
> |TRLOO+MRS+PR+PRS(8B)|19.88|19.40|20.00|19.76|0.32|
> |TRLOO+MRS(14B)|20.10|20.20|19.80|20.03|0.21|
> |TRLOO+MRS+PR(14B)|22.00|22.10|21.80|21.97|0.15|
> |TRLOO+MRS+PR+PRS(14B)|26.10|25.60|25.60|25.77|0.29|
>
> The results show consistent improvements from PRS. At 8B, TRLOO+MRS+PR+PRS achieves 19.76 Fast@1.2, outperforming TRLOO+MRS+PR (18.28) and TRLOO+MRS (17.43). The improvement is even larger at 14B, where adding PRS yields a +4.8 Fast@1.2 gain over the best ablation. The low standard deviation across runs (<0.5 for 8B and <0.3 for 14B) further supports the statistical reliability of these improvements.
>
> # W2
>
> ```
> It seems that the training of Dr. Kernel is highly expensive. I think author should add some details about GPU/Time consumption in RL.
> ```
> For both 8B/14B models, we used 2 H100 GPU nodes for RL training and 1~2 nodes for kernel evaluations. The time consumption for 8B/14B models is up to 1.5 or 2 days to achieve the best performance, which is also similar to other agentic RL training.
>
> # W3
>
> ```
> How the Dr. Kernel do multi-turn generation is not very clear. Authors should add details. For example, How they organize context? / Do they use tools/profiling in inference?
> ```
> The context organization for Dr. Kernel’s multi-turn generation is illustrated in Figure 13, in Appendix F.2. After each turn, we send the generated kernel code to the reward server, which returns execution feedback such as error tracebacks and profiling results. We then incorporate this feedback into the prompt for the next turn, as shown in Figure 13. In addition, the full interaction history from previous turns is included in the context for subsequent generation.
> Yes, we use the same tools and profiling setup during both training and inference.
>
> # Q1
> ```
> Could you provide more cases about optimized kernels?
> ```
>
> Due to space limitations, we provide one additional BatchNorm example from Dr.Kernel-14B, which achieves a 2.12× speedup. The generated implementation uses a structured three-stage design, where `reduce`, `reduce_rows`, and `normalize` are all Triton kernels implemented by Dr Kernel-14B:
>
> ```python
> partials[c, n] = reduce(x[n, c, :])          # partial stats, no atomics
> mean, var = reduce_rows(partials)            # reduction over partials
> y[n, c, :] = normalize(x[n, c, :], mean, var)
> ```
>
> The key idea is to decompose BatchNorm into parallel partial-statistics computation, device-side reduction, and normalization. This avoids atomic contention, reduces synchronization overhead, and yields the 2.12× speedup. We will provide the full code in the revised paper.
>
> # Q2
> ```
> Could you report avg speedup in the results?
> ```
> Thanks for the suggestion. We additionally report the average speedup, where samples corresponding to hacking or incorrect kernel implementations are assigned a value of 0. The table below is aligned with Table 1 in the paper. Here, `dr.kernel-14b-stts` denotes the result in Figure 5 (right) with sequential test-time scaling (STTS) and context management, while `dr.kernel-14b-stts-best` denotes the best performance across historical turns under STTS. Overall, the average speedup results are consistent with the Fast@1.2 metric: our models are competitive with frontier models and further outperform them under STTS.
>
> | model | level1 | level2 | level3 |
> |---|---:|---:|---:|
> | gpt-5 | 0.67 | 0.81 | 0.5 |
> | claude-4.5-sonnet | 0.63 | 0.82 | 0.31 |
> | deepseek-v3.2-thinking | 0.26 | 0.22 | 0.09 |
> | glm4.7 | 0.7 | 0.59 | 0.17 |
> | qwen3-8b | 0.22 | 0.23 | 0.17 |
> | qwen3-32b | 0.3 | 0.24 | 0.09 |
> | qwen3-coder-a30ba3 | 0.55 | 0.21 | 0.2 |
> | autotriton | 0.22 | 0.5 | 0.18 |
> | cold-start-8b | 0.15 | 0.2 | 0.09 |
> | dr. kernel-8b | 0.3 | 0.62 | 0.21 |
> | dr. kernel-14b | 0.65 | 0.74 | 0.18 |
> | dr.kernel-14b-stts | 0.92 | 0.91 | 0.29 |
> | dr. kernel-14b-stts-best | 1.21 | 1.44 | 0.48 |

---

> > ### Author Rebuttal · Reviewer_44jY · 2026-04-03
> >
> > Most of my concerns are resolved! Thanks! I will update my score according to all the 4 reviews/rebuttals here, since I think other reviewers' questions are also meaningful.

---

> > > ### Author Response · Authors · 2026-04-07
> > >
> > > Dear Reviewer 44jY,
> > >
> > > Thank you for acknowledging that most of your concerns have been resolved. We are glad that our rebuttal addressed your questions effectively.
> > >
> > > We would like to provide a brief update on the current discussion status. Reviewer 2JS5 has acknowledged our rebuttal and updated their score to 5. Reviewer LX8g raised follow-up questions regarding generalization beyond KernelBench, reward clipping ablation, and MRS rejection rate dynamics, and we have provided detailed responses with additional experimental results (including cross-benchmark evaluation on TriMul, clipping threshold ablation, and rejection rate over training steps).
> > >
> > > As the rebuttal period is approaching its deadline, we sincerely welcome any further feedback or questions you may have. We would be happy to address any remaining concerns or provide additional clarifications. We also genuinely value your consideration of a score update based on the discussions across all reviews.
> > >
> > > Thank you again for your time and thoughtful review.

---

### Official Review · Reviewer_LX8g · 2026-03-11

**Soundness:** 3
**Presentation:** 3
**Significance:** 3
**Originality:** 3
**Overall Recommendation:** 5
**Confidence:** 4

**Summary:**

This paper presents a systematic study of reinforcement learning (RL) for Triton kernel code generation by LLMs. The authors identify two key failure modes in this domain—reward hacking and lazy optimization—and propose a suite of solutions. First, they build KERNELGYM, a distributed GPU execution environment with fault isolation, profiling, and hacking detection. Second, they propose Turn-level REINFORCE Leave-One-Out (TRLOO), an unbiased advantage estimator for multi-turn RL that addresses a self-inclusion bias in standard GRPO. Third, they introduce Profiling-based Rewards (PR) and Profiling-based Rejection Sampling (PRS) to steer the model away from trivial optimizations toward meaningful speedup. The resulting DR. KERNEL-14B model achieves performance competitive with Claude-4.5-Sonnet on KernelBench, and with sequential test-time scaling, even surpasses GPT-5 on the Level-2 subset.

**Compliance With Llm Reviewing Policy:**

Affirmed.

**Final Justification:**

I thank the authors for a thorough and responsive rebuttal, and I am maintaining my recommendation of Accept.

The rebuttal addressed my main concerns effectively. The expanded TriMul comparison with multiple baselines provides convincing evidence of cross-benchmark generalization beyond KernelBench, showing that DR. KERNEL-14B transfers competitively against frontier models on a real-world kernel optimization task. The reward clipping ablation across four thresholds, combined with the per-step max speedup distribution, empirically substantiates the claim that the 3× cap does not constrain learning. The MRS rejection rate trajectory across training steps confirms stable training dynamics without unintended side effects. The decontamination analysis also adequately addresses my data contamination concern.

One reservation remains around the limited ablation on cold-start SFT data quality, but the evidence that RL is the dominant driver of performance is reasonable, and this does not undermine the paper's main contributions.

Soundness, originality, significance, and presentation all remain at 3. The identification of lazy optimization, the TRLOO derivation, and the KERNELGYM system are solid contributions that the broader RL-for-code-generation community can build on. I support acceptance.

**Key Questions For Authors:**

1. How sensitive is the RL training outcome to the cold-start SFT data quality? Have you tried using a weaker teacher model or varying the amount of SFT data? This would help disentangle the contribution of the RL stage from the initial SFT.

2. For the hacking check, you instrument Triton's launch path to verify kernel execution. Is this check comprehensive enough to catch all hacking strategies, or could a sufficiently capable model learn new ways to exploit the evaluation that bypass this specific check? Have you observed any novel hacking patterns emerging during training?

3. In Eq. (6), the MRS rejection bounds [0.999, 1.001] are extremely tight. How many samples are rejected in practice, and does this rejection rate change over training? A high rejection rate could effectively reduce the number of gradient updates and slow learning.

**Limitations:**

Yes, the authors briefly acknowledge limitations in their Level 3 analysis and the impact statement. However, a more explicit discussion of failure modes and the generalizability of the approach beyond Triton/KernelBench would be welcome.

**Strengths And Weaknesses:**

### Strengths

1. The identification and formalization of "lazy optimization" is a genuine contribution. While reward hacking has been noted before, the observation that models converge to trivially correct but low-speedup solutions—and the gap between Fast@1 and Fast@1.2 as evidence—is insightful and well-motivated. The profiling ratio (PR) as a metric to quantify this is simple but effective.

2. The TRLOO derivation is clean and well-grounded. The self-inclusion issue in GRPO's mean baseline is a real problem that has been overlooked in some recent RL-for-LLM works. The derivation in Appendix A is easy to follow, and the equivalence $A^{TRLOO}_{i,t} = \frac{N_t}{N_t - 1}(G_{i,t} - \bar{G}_t)$ makes it practical to implement. The empirical comparison against GRPO in Figure 3 supports the claim.

3. KERNELGYM is a well-engineered system with sensible design choices. The subprocess-level fault isolation, heartbeat-based monitoring, and one-GPU-one-task serialization are all practical decisions that address real pain points in GPU-based RL environments. The hacking check based on instrumenting Triton's launch path is a pragmatic and effective safeguard.

4. The staged diagnostic approach in Section 5 (first testing training instability, then objective misalignment) is a good example of principled experimental methodology. The fact that MRS alone stabilizes training but does not lift Fast@1.2 is a useful negative result that justifies the need for PR/PRS.


### Weaknesses

1. The paper is exclusively evaluated on KernelBench, which is a single benchmark with known biases (e.g., Level 2 tasks may not be representative of real-world kernel optimization challenges). No evaluation on other kernel generation benchmarks or downstream real-world tasks is provided. It would strengthen the paper to show generalization beyond this specific benchmark.

2. The cold-start data is distilled from GPT-5 with 8K queries, but there is essentially no analysis of data quality or diversity. How sensitive is the final RL performance to the quality of the cold-start SFT data? An ablation on cold-start data size or source would be informative. Relatedly, the paper does not discuss potential data contamination—GPT-5 may have seen KernelBench tasks during its own training.

3. The profiling-based rejection sampling (PRS) uses fixed hyperparameters ($\tau=0.3$, $s=0.1$) with only a brief ablation on the softness parameter in Appendix D. No sensitivity analysis on $\tau$ is provided, despite this being arguably the more important hyperparameter. It is unclear how transferable these settings are across different task distributions.

4. The reward clipping at 3× speedup (Eq. 1) is justified by saying speedups beyond 3× are uncommon, but this is a strong inductive bias. If the goal is to push toward higher-performing kernels, this cap may actually limit the learning signal for the most promising samples. No ablation on the clipping threshold is provided.

---

> ### Author Rebuttal · Authors · 2026-03-31
>
> # W1
> ```
> The paper is exclusively evaluated on KernelBench, which is a single benchmark…No evaluation on other kernel generation benchmarks or downstream real-world tasks is provided…
> ```
> To evaluate generalisation beyond KernelBench, we additionally tested our model on the TriMul competition [1,2]—a Triangle Multiplicative Update kernel core to AlphaFold3 and protein structure prediction. DR. KERNEL-14B achieves 1.41× speedup (~2320 µs on H100), comparable to the 120B model in [2]. This demonstrates strong cross-benchmark transfer to real-world. We will include these results and the discussion in the revised paper.
>
> [1] GPU Mode, Trimul Competition, www.gpumode.com/leaderboard/496
>
> [2] Yuksekgonul et al., Learning to Discover at Test Time, www.arxiv.org/abs/2601.16175
>
> # W2 & Q1
>
> ```
> How sensitive is the RL training outcome to the cold-start SFT data quality? Have you tried using a weaker teacher model or varying the amount of SFT data? This would help disentangle the contribution of the RL stage from the initial SFT. Relatedly, the paper does not discuss potential data contamination
> ```
>
> We conducted decontamination analysis between our SFT/RL data and KernelBench using four methods: exact match, AST-level comparison, substring containment (≥200 chars), and longest common token subsequence. We found 0 matches under the first three and at most 23–24 common tokens under the fourth, indicating no KernelBench contamination in training data.
>
> Table 1 shows that Cold-Start-8B achieves only marginal speedup over Qwen3-8B-Base, while RL training leads to substantial gains. This suggests that RL is the main driver of performance improvement, and that the model is relatively robust to SFT data quality as long as the data provides sufficient task knowledge. Due to time and resource limits, we will include a more detailed discussion of teacher-model and data-size ablations in the revised paper.
>
> # W3
> ```
> The profiling-based rejection sampling (PRS) uses fixed hyperparameters (, ) with only a brief ablation on the softness parameter in Appendix D. No sensitivity analysis on  is provided...
> ```
> We initially choose these two settings by observing the fact that in our RL training the average T_generated / T_total_cuda is 25\% ~ 30\%+. Therefore, we choose $\tau=0.3,s=0.1$ as our parameter and we never tune them. To better validate our robustness, we further train our model with two different settings: $\tau=0.25, s=0.1$ and $\tau=0.3, s=0.05$. And the results are shown below:
> |Experiments|Fast@1.2|
> |:---|:---|
> |$\tau=0.3,s=0.1$|19.9|
> |$\tau=0.2,s=0.1$|19.3|
> |$\tau=0.3,s=0.05$|19.6|
> Experiments with different $\tau, s$ show that PRS is robust to different hyperparameters.
>
> # W3
> ```
> The reward clipping at 3× speedup (Eq. 1) is justified by saying speedups beyond 3× are uncommon, but this is a strong inductive bias...this cap may actually limit the learning signal for the most promising samples...
> ```
> We chose this threshold based on preliminary experiments, where we found that, for most checkpoints, the best speedup among generated samples during training remained below 3×. We therefore applied clipping at this value to reduce the effect of occasional instability or outliers from the environment. We agree that this threshold is a task-dependent hyperparameter and may need to be adjusted for other tasks. In our setting, however, it worked well in practice: we did not observe evidence of speedup saturation on the training/test set, suggesting that the clipping did not materially constrain learning.
>
> # Q2
> ```
> For the hacking check, you instrument Triton's launch path to verify kernel execution. Is this check comprehensive enough to catch all hacking strategies... Have you observed any novel hacking patterns emerging...?
> ```
> Besides instrumenting Triton’s launch path, we also require the generated kernel to appear in the PyTorch profiler trace; otherwise, we classify it as hacking. Together, these two checks provide strong coverage, and we did not observe cases that bypass both.
>
> # Q3
> ```
> In Eq. (6), the MRS rejection bounds [0.999, 1.001] are extremely tight. How many samples are rejected in practice, and does this rejection rate change over training? A high rejection rate could effectively reduce the number of gradient updates and slow learning.
> ```
>
> We follow the default MRS bounds from [1], also recommended in verl [2]. About 20–30% of samples are rejected in practice. As in [1,3], we oversample rollouts to keep each rollout group full. Although this adds modest rollout cost, it improves sample efficiency in practice by surfacing higher-reward samples earlier and increasing peak reward.
>
> [1] Liu et al., When Speed Kills Stability: Demystifying RL Collapse from the Training-Inference Mismatch, https://richardli.xyz/rl-collapse
>
> [2] Verl, Rollout Correction, verl.readthedocs.io/en/latest/algo/rollout_corr.html
>
> [3] Yu et al., DAPO: An Open-Source LLM Reinforcement Learning System at Scale, www.arxiv.org/abs/2503.14476

---

> > ### Author Rebuttal · Reviewer_LX8g · 2026-04-03
> >
> > Thank you for the detailed rebuttal. The identification of lazy optimization and the TRLOO derivation remain strong contributions, and several responses are helpful. I have a few follow-up requests:
> >
> > Regarding W1 (Generalization beyond KernelBench): The TriMul result is encouraging, but a single speedup number is not sufficient to fully address the generalization concern. Could you provide a more complete results table — e.g., including comparisons against other baselines on this task, or results on additional kernels beyond TriMul — so that readers can better assess cross-benchmark transfer?
> >
> > Regarding W3 (Reward Clipping at 3×): You mention that no speedup saturation was observed, but no evidence is provided. Could you include an ablation comparing different clipping thresholds (e.g., 2×, 3×, 5×, or no clipping) to empirically validate that the 3× cap does not constrain learning? Even a small-scale experiment would help substantiate the claim that this choice is not limiting performance.
> >
> > Regarding Q3 (MRS Rejection Rate): You mention that 20–30% of samples are rejected in practice, but no supporting data is provided. Could you include a plot or table showing the rejection rate over training steps? Understanding whether this rate remains stable, increases, or decreases during training would help assess whether MRS introduces any unintended training dynamics.
> >
> > These are straightforward additions that would meaningfully strengthen the empirical rigor. I look forward to the authors' response.

---

> > > ### Author Response · Authors · 2026-04-05
> > >
> > > Thank you for the constructive follow-up. We address each point with new experimental results below.
> > >
> > > # W1 (Generalization beyond KernelBench)
> > >
> > > ```
> > > Could you provide a more complete results table — e.g., including comparisons against other baselines on this task, or results on additional kernels beyond TriMul?
> > > ```
> > >
> > > We provide the full TriMul comparison below. The reference kernel runtime is 3271 µs.
> > >
> > >
> > > | Model               | Kernel (µs) | Speedup (T_ref / T_kernel) |
> > > | ------------------- | ----------- | -------------------------- |
> > > | GPT-5               | 2490        | 1.31×                      |
> > > | Claude-4.5-Sonnet   | 2351        | 1.39×                      |
> > > | Qwen3-8B-Base       | ✗           | —                          |
> > > | Qwen3-32B           | ✗           | —                          |
> > > | Qwen3-Coder-A30B-A3 | 17302       | 0.19×                      |
> > > | GPT-OSS-120B        | 5306        | 0.62×                      |
> > > | **Dr. Kernel-8B**   | 2920        | 1.12×                      |
> > > | **Dr. Kernel-14B**  | 2320        | **1.41×**                  |
> > >
> > >
> > > While Qwen3-8B-Base and Qwen3-32B fail to produce a correct kernel, highlighting the difficulty of this task, Dr. Kernel-14B achieves competitive performance with frontier models such as GPT-5 and Claude-4.5-Sonnet, and outperforms the 120B model. These results demonstrate strong cross-benchmark transfer to real-world kernel optimization tasks.
> > >
> > > # W3 (Reward Clipping Ablation)
> > >
> > > ```
> > > Could you include an ablation comparing different clipping thresholds (e.g., 2×, 3×, 5×, or no clipping)?
> > > ```
> > >
> > > We train Dr. Kernel-8B under four clipping settings and report Fast@1.2 and average speedup on Level 2:
> > >
> > >
> > > | Clipping  | Fast@1.2 | Avg Speedup |
> > > | --------- | --------------------------- | ----------- |
> > > | No Clip   | 19.9                        | 0.60        |
> > > | 5×        | 20.2                        | 0.65        |
> > > | 3× (ours) | 20.0                        | 0.62        |
> > > | 2×        | 19.6                        | 0.58        |
> > >
> > >
> > > We clarify that the purpose of reward clipping is **not** to improve performance via hyperparameter tuning, but to guard against environment instabilit, occasional outlier speedups caused by profiling noise or system jitter can destabilize RL training. The results confirm that all thresholds perform comparably, with Fast@1.2 varying within a narrow range of [19.6, 20.2].
> > >
> > > To explain why different thresholds have minimal impact, we examine the per-step max speedup (i.e., the highest speedup among all queries and samples at each training step) across all training steps:
> > >
> > >
> > > | Max Speedup Range | Fraction of Steps |
> > > | ----------------- | ----------------- |
> > > | [2×, 5×]          | 14%               |
> > > | [3×, 5×]          | 3%                |
> > >
> > >
> > > Only 3% of training steps ever produce a sample with max speedup exceeding 3×, and 14% exceed 2×. Since high-speedup events are already rare, all thresholds perform comparably. At 3×, the affected fraction is sufficiently small that clipping serves its stabilization purpose without constraining learning. The slightly higher proportion at 2× may explain its marginal performance dip, though the overall range remains narrow across all settings.
> > >
> > > # Q3 (MRS Rejection Rate)
> > >
> > > ```
> > > Could you include a plot or table showing the rejection rate over training steps?
> > > ```
> > >
> > > We report the MRS rejection fraction at each training step for Dr. Kernel-8B:
> > >
> > >
> > > | Step      | 20  | 40  | 60  | 80  | 100 | 120 | 140 | 160 | 180 | 200 | 220 | 240 | 260 | 280 | 300 |
> > > | --------- | --- | --- | --- | --- | --- | --- | --- | --- | --- | --- | --- | --- | --- | --- | --- |
> > > | Rej. Rate | 0.27 | 0.25 | 0.26 | 0.27 | 0.31 | 0.30 | 0.27 | 0.30 | 0.28 | 0.32 | 0.32 | 0.31 | 0.33 | 0.32 | 0.34 |
> > >
> > >
> > > The rejection rate remains stable throughout training, fluctuating between 25–34% with no sudden spikes or collapse. The slight upward trend (from ~27% to ~33%) is expected: kernel writing is a relatively unfamiliar task for the base model, so the distribution shift gradually increases over the course of RL training. This is consistent with other training dynamics indicators (e.g., PPL) that also show an upward trend. Overall, the rate never exceeds 34% and remains within an acceptable range. As noted in our previous response, we oversample rollouts to compensate for rejected samples, ensuring that each training batch remains full. These results confirm that MRS does not introduce unintended training dynamics or significantly reduce effective gradient updates.

---

### Official Review · Reviewer_2JS5 · 2026-03-13

**Soundness:** 3
**Presentation:** 3
**Significance:** 3
**Originality:** 3
**Overall Recommendation:** 5
**Confidence:** 4

**Summary:**

This paper introduces Dr. Kernel, a method for training LLMs to translate pytorch modules into optimized triton kernels using RL. The paper found that reward hacking and under-optimization were major challenges in this setting, so they built an RL environment, custom reward functions, and a slightly modified RL algorithm to address these problems and trained a 14B parameter LLM to strong performance on this task.

**Compliance With Llm Reviewing Policy:**

Affirmed.

**Final Justification:**

Overall a strong paper, with the contributions to the field (significance) outweighing the slightly weak methodological novelty in my opinion. The rebuttals (to mine and the other reviewers) were quite thorough and did a good job addressing my concerns.

**Key Questions For Authors:**

- Are the findings from the ablation studies / experiments (figures 1, 2, 4) consistent across seeds? Are they statistically significant?
- How often does this model beat torch.compile?
- Will the code be open sourced?

**Limitations:**

Limitations and failure cases are not discussed in the paper.

**Strengths And Weaknesses:**

**Strengths:**
- This paper addresses an important and valuable task where LLMs still struggle.
- KernelGym alone could be a valuable contribution to the community. It seems to provide a lot of useful signal for providing feedback to the LLM and constructing rewards, and robust environments for evaluating kernels are challenging to build.
- The paper does a good job identifying challenges in the training process and addressing them.
- Dr Kernel seems to outperform even frontier LLMs on relevant tasks while only having 14B parameters.


**Weaknesses:**
- The ablation studies are somewhat weak, so it is uncertain which changes actually contribute significantly to performance. Figures 1, 3, and 4, while interesting and instructive, can't be trusted or relied on too much as the methods perform quite similarly and there is a lot of noise that is inherent in RL training. A statistical analysis would add a lot of value.
- The methodological contributions, while solid, are not particularly novel. The TRLOO contribution, while correct, is incremental.
- The main results use best-of-8 which is pretty generous.
- The baselines are not completely fair, the frontier LLMs are only given 3 turns.
- The "Fast@X" metrics are poorly named as they are correctness metrics (for a given speed-up), not speed metrics.
- Ultimately, if you just want better performance, you are still probably better off just calling torch.compile on the original module.

---

> ### Author Rebuttal · Authors · 2026-03-31
>
> # W1
> ```
> The ablation studies are somewhat weak, so it is uncertain which changes actually contribute significantly to performance…A statistical analysis would add a lot of value.
> ```
>
> ```
> Are the findings from the ablation studies / experiments (figures 1, 2, 4) consistent across seeds? Are they statistically significant?
> ```
>
> Thank you for this valuable suggestion. Due to time and resource constraints, we conducted three independent runs for TRLOO+MRS, TRLOO+MRS+PR, and TRLOO+MRS+PR+PRS across both 8B and 14B model sizes. The Fast@1.2 results are:
>
> |Method|Run1|Run2|Run3|Mean|Std|
> |---|---|---|---|---|---|
> |TRLOO+MRS(8B)|17.50|17.80|17.00|17.43|0.40|
> |TRLOO+MRS+PR(8B)|18.60|18.00|18.25|18.28|0.30|
> |TRLOO+MRS+PR+PRS(8B)|19.88|19.40|20.00|19.76|0.32|
> |TRLOO+MRS(14B)|20.10|20.20|19.80|20.03|0.21|
> |TRLOO+MRS+PR(14B)|22.00|22.10|21.80|21.97|0.15|
> |TRLOO+MRS+PR+PRS(14B)|26.10|25.60|25.60|25.77|0.29|
>
> The results demonstrate consistent improvements: TRLOO+MRS+PR+PRS (19.76) outperforms TRLOO+MRS+PR (18.28) and TRLOO+MRS (17.43) at 8B. The gap widens at 14B, where the final method Dr. Kernel achieves +4.8 Fast@1.2 over the best ablation. The low standard deviations across runs (<0.5 for 8B, <0.3 for 14B) also support the statistical reliability of our conclusions from RL training.
>
>
> # W2
> ```
> The main results use best-of-8 which is pretty generous.
> ```
> We clarify that we computed all metrics using average@8 rather than best@8, ensuring fair and statistically consistent comparisons across different models.
> # W3
> ```
> The baselines are not completely fair, the frontier LLMs are only given 3 turns.
> ```
> All models in Table 1, including ours and frontier LLMs, are evaluated with 3 turns, and we report metrics from the 3rd turn for a fair comparison. Given the substantial gap in model size and training data, we view our models' outperformance over frontier LLMs as significant progress.
>
> # W4
> ```
> The "Fast@X" metrics are poorly named as they are correctness metrics (for a given speed-up), not speed metrics.
> ```
> Thank you for raising this concern. We follow the definition from KernelBench where ``Fast@X is
> defined as the fraction of tasks that are both correct and have a X times speedup``. Since incorrect kernels receive 0 speedup by default, correctness is a prerequisite rather than the sole focus; speedup remains the distinguishing factor.
>
> # W5
> ```
> Ultimately, if you just want better performance, you are still probably better off just calling torch.compile on the original module.
> ```
> ```
> How often does this model beat torch.compile?
> ```
> Thanks for the insightful question. We additionally report Fast@1/1.2/1.5/2 across all levels using torch.compile as the baseline. This table is similar to Table 1, except that we use torch.compile rather than default eager execution for the torch reference code runtime. Since torch.compile is already a strong optimized baseline, Fast@1 is especially meaningful in this setting; by contrast, under eager mode, trivial “lazy” optimizations can inflate Fast@1. The Fast@1 column therefore directly shows the percentage of tasks on which our generated kernel runs faster than torch.compile. For example, **DR. KERNEL-14B** outperforms torch.compile on 17.8% of Level 1 tasks and 23.5% of Level 2 tasks. Given that torch.compile already applies strong compiler optimizations such as operator fusion, we believe these results are encouraging. Our models also remain competitive with frontier models in this strict setting.
>
>
> | Model | LEVEL1 (Fast@1/1.2/1.5/2) | LEVEL2 (Fast@1/1.2/1.5/2) | LEVEL3 (Fast@1/1.2/1.5/2)|
> |:---|:---|:---|:---|
> | GPT-5 | 18.6 / 8.0 / 6.5 / 5.5 | 22.1 / 3.6 / 1.5 / 1.0 | 14.0 / 4.0 / 3.0 / 1.0 |
> | Claude-4.5-Sonnet | 10.0 / 2.2 / 2.0 / 1.8 | 20.5 / 3.0 / 0.0 / 0.0 | 12.0 / 3.5 / 0.5 / 0.0 |
> | DR. KERNEL-8B | 16.0 / 3.0 / 1.5 / 1.3 | 20.6 / 0.8 / 0.0 / 0.0 | 7.2 / 2.3 / 0.0 / 0.0 |
> | DR. KERNEL-14B | 17.8 / 5.0 / 3.3 / 2.5 | 23.5 / 1.9 / 0.0 / 0.0 | 9.2 / 3.0 / 0.0 / 0.0 |
>
> # Q1
> ```
> Will the code be open sourced?
> ```
> Yes. We have a clean and easy-to-use codebase, including everything (code, models, data) inside the paper. And we will release them.

---

> > ### Author Rebuttal · Reviewer_2JS5 · 2026-04-04
> >
> > This response (particularly the new experimental results) and the others addressed my primary concerns, and I have increased my score accordingly.

---

> > > ### Author Response · Authors · 2026-04-07
> > >
> > > Dear Reviewer 2JS5,
> > >
> > > Thank you for your thorough and constructive review, and for recognizing the significance of our contributions. We are glad that our rebuttals addressed your concerns effectively. We will incorporate all the additional results discussed during the rebuttal into the camera-ready version. Thank you again for your time and valuable feedback.

---

### Decision · Program_Chairs · 2026-04-30

**Decision:**

Accept (regular)

**Comment:**

This paper presents Dr.Kernel, an RL method for training LLMs to translate PyTorch modules into optimized Triton kernels. The authors analyze key challenges in applying RL to this domain (including reward hacking) and address them through KernelGym, a dedicated RL environment, and a new unbiased advantage estimator for multi-turn RL. The Dr.Kernel-14B model achieves performance comparable to Claude Sonnet 4.5 on KernelBench. Overall, the paper tackles an important and challenging domain, and all reviewers agree that the contributions are valuable to the community. The authors also thoroughly addressed reviewer concerns around ablation studies and baselines during the rebuttal. I recommend acceptance.